# Revealing Multimodal Causality with Large Language Models

**Jin Li**[1], **Shoujin Wang**[1]*, **Qi Zhang**[2], **Feng Liu**[3], **Tongliang Liu**[4],
**Longbing Cao**[5], **Shui Yu**[1], **Fang Chen**[1]
[1]University of Technology Sydney  [2]Tongji University
[3]University of Melbourne  [4]University of Sydney  [5]Macquarie University
jin.li-4@student.uts.edu.au {shoujin.wang,shui.yu,fang.chen}@uts.edu.au
zhangqi_cs@tongji.edu.cn feng.liu1@unimelb.edu.au
tongliang.liu@sydney.edu.au longbing.cao@mq.edu.au

## Abstract

Uncovering cause-and-effect mechanisms from data is fundamental to scientific progress. While large language models (LLMs) show promise for enhancing causal discovery (CD) from unstructured data, their application to the increasingly prevalent multimodal setting remains a critical challenge. Even with the advent of multimodal LLMs (MLLMs), their efficacy in multimodal CD is hindered by two primary limitations: (1) difficulty in exploring intra- and inter-modal interactions for comprehensive causal variable identification; and (2) insufficiency to handle structural ambiguities with purely observational data. To address these challenges, we propose MLLM-CD, a novel framework for multimodal causal discovery from unstructured data. It consists of three key components: (1) a novel contrastive factor discovery module to identify genuine multimodal factors based on the interactions explored from contrastive sample pairs; (2) a statistical causal structure discovery module to infer causal relationships among discovered factors; and (3) an iterative multimodal counterfactual reasoning module to refine the discovery outcomes iteratively by incorporating the world knowledge and reasoning capabilities of MLLMs. Extensive experiments on both synthetic and real-world datasets demonstrate the effectiveness of the proposed MLLM-CD in revealing genuine factors and causal relationships among them from multimodal unstructured data. The implementation code and data are available at `https://github.com/JinLi-i/MLLM-CD`.

## 1 Introduction

Causal discovery (CD), which aims to infer the underlying causal structures from the data, is a fundamental task across numerous real-world scenarios, including healthcare [1, 2], finance [3], and machine perception [4–6], playing a vital role in advancing scientific inquiry and human cognition. As data generation and collection technologies [7–9] continue to evolve, real-world scenarios are increasingly characterized by *multimodal* and *unstructured* data, such as the combination of texts, images, and/or audio, making CD particularly crucial yet challenging in practice. Effectively tackling CD in such cases typically involves two key steps: (1) identifying potential causal variables (factor discovery) and their values from the raw unstructured data, and (2) inferring the causal relationships among these variables (structure discovery). Traditional CD methods [10–12], though successful on structure discovery [13], heavily rely on predefined causal variables in structured data and lack the

---

*Corresponding author.

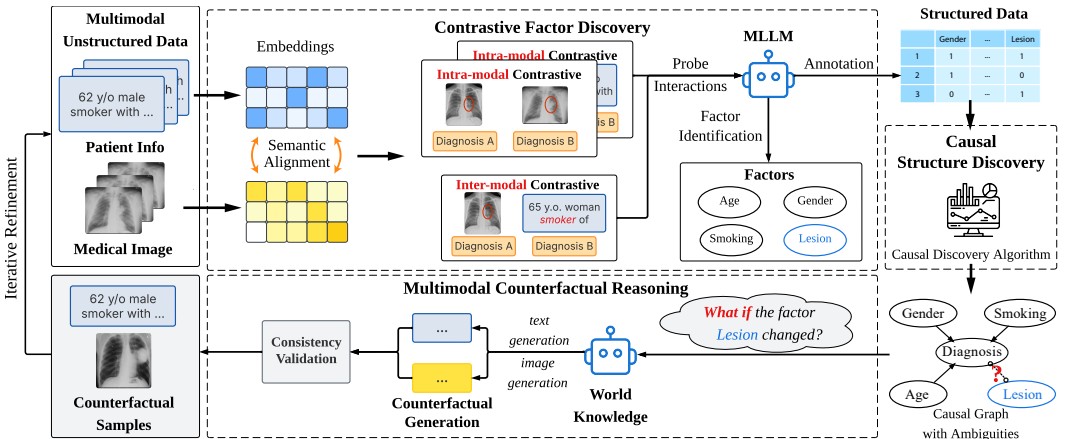

Figure 1: Illustration of MLLM-CD in lung cancer diagnosis. It first employs contrastive factor discovery with MLLMs to identify potential causal variables and form structured data. Then, a CD algorithm is performed to infer causal structures. To further reduce ambiguities, it leverages MLLM's world knowledge to generate multimodal counterfactual samples for iterative refinement.

inherent capability to perform factor discovery. As such, they are ill-equipped to handle unstructured data that lacks predefined variables, let alone the more complex multimodal unstructured data.

Identifying genuine causal variables from unstructured data is challenging and costly, as it often requires extensive manual feature engineering and domain expertise [14]. This makes it difficult to scale with the increasing volume and complexity of modern data. Recently, the advent of large language models (LLMs) [15–19] offers a promising opportunity to bridge this gap. With their remarkable capabilities in understanding context and processing natural language, LLMs can be leveraged to automate factor discovery from raw textual data [20, 21], paving the way for the following structure discovery. Integrating LLMs into CD pipelines holds significant potential to broaden the scope and applicability of conventional CD, yet this direction remains largely under-explored.

To the best of our knowledge, COAT [20] is the only existing work that explicitly targets CD from unstructured data using LLMs, marking a significant step forward in this direction. It first extracts causal variables via LLMs and then infers the causal relationships with traditional CD methods. However, it is limited to unimodal data, typically text. In contrast, the rapid proliferation of multimedia content necessitates CD methods capable of effectively handling multimodal information [22, 23] across various domains, such as medical diagnosis [24], where clinical notes, medical images, and lab results collectively inform causal understanding [25–27]. While it may seem feasible to adapt unimodal approaches like COAT to multimodal scenarios using multimodal LLMs (MLLMs) [28], our findings suggest that such a naive adaptation falls short. As shown in Figure 2, the adapted COAT struggles with two critical limitations on multimodal unstructured data: (1) it uncovers only a small set of causal factors, and (2) the inferred causal edges remain undirected.

Such limitations stem from two key challenges (CHs) inherent to multimodal CD which can not be addressed by simple adaptations. **CH1. Probing Intra- and Inter-modal Interactions**. Causal variables are often embedded in different modalities, and become identifiable only when interactions between them are probed (e.g., the observed interaction "smaller apples (image) have lower target scores (text)" helps reveal the variable of `size`). Without guidance from such cross-modal interactions, existing methods tend to identify only the most salient factors (e.g., `taste`, `aroma`, and `defects`), while overlooking more implicit yet equally important ones (e.g., `nutrition`). **CH2. Handling Structural Ambiguities**. Inferring accu-

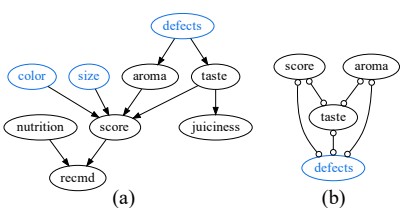

Figure 2: Results on MAG dataset[1]. (a) Ground truth. (b) Causal graph discovered by COAT. Visual and verbal causal variables are presented in blue and black.

---

[1]Results are based on Gemini 2.0. Please see Section 4.1 and Appendix D.3 for detailed experimental settings.

rate causal structures from observational data alone is inherently challenging [29], as multiple causal structures can yield the same statistical dependencies [13], leading to undesired ambiguity. This issue is further amplified in multimodal settings since they typically involve more causal variables while the number of possible causal structures grows exponentially with the number of variables.

To overcome these pressing challenges, we introduce MLLM-CD, a novel framework that integrates the strengths of MLLMs with conventional CD methods. As illustrated in Figure 1, MLLM-CD is devised with three key modules. First, a *contrastive factor discovery* module is introduced for multimodal factor identification. It encourages the MLLM to explore intra- and inter-modal interactions through variations in semantically contrastive sample pairs, helping reveal implicit variables hidden in multimodal inputs. Second, *causal structure discovery* is employed to infer the causal structure among the identified factors. Third, a novel *iterative multimodal counterfactual reasoning* module is proposed to refine the inferred causal structures by reducing structural ambiguities. Specifically, by performing counterfactual reasoning on factors with uncertain relationships, it generates hypothetical yet causally consistent multimodal samples for iterative refinement. In this way, MLLMs offer counterfactual evidence beyond the observational data to mitigate structural ambiguities through their world knowledge and reasoning capabilities. Our main contributions are summarized as follows:

- We propose MLLM-CD, to the best of our knowledge, the first work for causal discovery in multimodal unstructured data with a novel framework that integrates MLLMs into CD pipelines. This significantly extends the scope of CD beyond structured and/or unimodal data settings for practical use.

- We introduce an innovative contrastive factor discovery module that encourages MLLMs to capture the intra- and inter-modal interactions for accurate factor identification under the multimodal unstructured context.

- We design a novel iterative multimodal counterfactual reasoning module to iteratively refine the inferred causal structures by generating multimodal counterfactuals.

- We establish the first benchmark work for multimodal unstructured causal discovery with synthetic and real-world datasets. Extensive experiments demonstrate the superior performance of MLLM-CD in both factor discovery and causal structure discovery.

## 2   Related Work

**Causal Discovery** [11, 30] aims to uncover the graphical structures, e.g., directed acyclic graph (DAG), with a causal interpretation from observational data. Traditional causal discovery methods largely fall into two categories: constraint-based [31–34] and score-based approaches [35–37]. Constraint-based methods, such as PC [31] and FCI [31, 32], learn the causal graph structure by performing a series of conditional independence tests on the data. They rely on assumptions like the causal Markov condition and faithfulness [13] to link conditional independence relations to the graph structure. Score-based methods, like GES [38] and NOTEARS [36], search for a graph structure that maximizes a defined scoring function (e.g., Bayesian Information Criterion [39]). While powerful under ideal conditions, these methods face challenges in real-world applications, where data is typically unstructured and multimodal. Our work builds upon these foundational statistical CD methods and addresses their inherent limitations in multimodal and unstructured contexts by integrating the multimodal understanding and reasoning capabilities of MLLMs.

**LLM for Causal Discovery** [40–47] has recently shown great potential in supporting conventional CD pipelines. One of the main approaches is to leverage the general and domain-specific knowledge of LLMs to refine the discovery results [48–58] from statistical CD methods. However, as a post-refinement of conventional CD methods, these approaches also suffer from limitations in unstructured and multimodal data. Motivated by the advanced capabilities of LLMs in understanding natural language descriptions, another line of work focuses on using LLMs to identify causal factors [20] and structures [29, 59–61] from unstructured data. For instance, Liu *et al.* [20] introduce a LLM-driven CD framework, which uses LLMs to identify the Markov blanket of the target variable from natural language descriptions. To test the effectiveness of LLMs in causal relationship discovery, Kiciman *et al.* [29] propose a pairwise prompting method and find that LLMs can be a good candidate for identifying relationships between a pair of variables. Vashishtha *et al.* [60] further extend this idea to a more accurate and robust triplet-based prompting to resolve the issue of multiple cycles. Nevertheless, these methods are not devised to handle the multimodal nature of data, thus limiting their effectiveness and applicability. Moreover, integrating the contextual understanding and reasoning capabilities

of LLMs into statistical CD pipelines—without compromising their statistical rigor—remains a non-trivial challenge. These limitations motivate the development of our proposed approach.

**Causal Representation Learning (CRL)** [62–64] aims to extract high-level representations and causal dependencies from low-level observational data, such as images or videos. Much of the work in this area is related to disentangled representation learning [65–68], which tries to map distinct factors of variation in the data to separate latent variables. In addition to observational data, using data generated under interventions or multiple environments has also been explored to learn more robust and causally meaningful representations [69–71]. While CRL has shown promise in providing provable identifiability of latent causal variables under certain assumptions, it remains a developing field with open challenges [69, 72, 73] in practical use. In this paper, we aim to identify human-interpretable causal variables and corresponding causal structures from multimodal unstructured data by leveraging the contextual understanding and reasoning capabilities of MLLMs.

# 3 Multimodal Causal Discovery

## 3.1 Problem Definition

We consider a dataset $\mathcal{D} = \{\mathbf{X}_1, \mathbf{X}_2, \cdots, \mathbf{X}_n\}$, where each sample $\mathbf{X}_k \in \mathcal{X}$ comprises observations from multiple modalities. Let $\mathbf{X}_k = (\mathbf{x}_{k1}, \mathbf{x}_{k2}, \cdots, \mathbf{x}_{km})$, where $\mathbf{x}_{kj}$ represents the $j$-th modality (e.g., text, image, video) of the $k$-th sample. We assume there exists a set of true, unobserved potential factors (i.e., causal variables) $\mathbf{V}^* = \{V_1^*, V_2^*, \cdots, V_{d'}^*\}$ that, along with a target variable $Y$, govern the data generation process. The true causal relationships among $\mathbf{V}^* \cup \{Y\}$ are represented by an unknown true causal directed acyclic graph (DAG) $\mathcal{G}^* = (\mathbf{V}^* \cup \{Y\}, \mathbf{E}^*)$. To discover the true causal graph, two main tasks need to be accomplished: (1) identifying potential factors from multimodal unstructured data, and (2) inferring the causal structure among these factors as a causal graph.

**Factor Identification.** This task involves identifying the relevant set of causal variables $\mathbf{V}$ from multimodal unstructured data $\mathcal{D}$. Crucially, it also requires annotating the value $v_{kj}$ for each variable $V_j \in \mathbf{V}$ corresponding to each sample $\mathbf{X}_k \in \mathcal{D}$. This process essentially transforms the unstructured multimodal data $\mathcal{D} = \{(\mathbf{x}_{k1}, \cdots, \mathbf{x}_{km})\}_{k=1}^n$ into a structured dataset $\mathcal{D}_{\mathrm{S}} = \{(v_{k1}, \cdots, v_{kd})\}_{k=1}^n$. Moving beyond solely unimodal factor discovery [20], this step must consider the multimodal nature of the data, and handle: (1) **Intra-modal complexity** to extract high-level semantic concepts within each modality (e.g., identifying the lesion in a medical image); (2) **Inter-modal dependencies** to recognize how concepts span or relate across different modalities (e.g., linking the lesion in medical images to the examination findings in clinical notes).

**Causal Structure Discovery.** Given the identified variables $\mathbf{V}$ and their annotated values in $\mathcal{D}_{\mathrm{S}}$, the goal is to infer causal relationships among $\mathbf{V}$ and the target variable $Y$, forming a causal graph $\mathcal{G} = (\mathbf{V} \cup \{Y\}, \mathbf{E})$ that best explains the statistical dependencies observed in $\mathcal{D}_{\mathrm{S}}$. However, even with the identified variables, this task remains challenging due to issues like Markov equivalence classes [13], where multiple DAGs encode the same conditional independencies.

**Our proposed MLLM-CD framework** aims to address the challenges in these tasks by leveraging the advanced understanding and reasoning capabilities of MLLMs. It consists of three key components: (1) a contrastive factor discovery (CFD) module, which probes intra- and inter-modal interactions using contrastive signals, facilitating accurate factor identification; (2) a causal structure discovery module to infer causal structures based on statistical dependencies; and (3) an iterative multimodal counterfactual reasoning (MCR) module to iteratively refine the CD and reduce ambiguities by incorporating both observational data and counterfactual samples generated from the world knowledge of MLLMs. The overall framework is illustrated in Figure 1.

## 3.2 Contrastive Factor Discovery

We propose a CFD module to identify potential causal factors $\mathbf{V}^{(t)}$ and annotate corresponding values in the $t$-th round. This module adopts an MLLM, denoted as $\Psi$, to explore the intra- and inter-modal interactions with contrastive signals, from which it can identify the underlying factors.

**Semantic Representation:** We begin by obtaining semantically aligned multimodal representations for each sample. As a common practice [74], we utilize pre-trained multimodal foundation models, such as CLIP [75], known for their ability to extract consistent embeddings from heterogeneous modalities. By adopting appropriate foundation models [76–78] as $f_i(\cdot)$, we can effectively extract semantic representations for the $k$-th sample in $i$-th modality $\mathbf{e}_{ki} = f_i(\mathbf{x}_{ki})$.

**Intra-modal Contrastive Exploration:** To uncover factors embedded within $i$-th modality, we select the top-$K$ pairs of samples $\mathcal{P}_i = \{(\mathbf{x}_{ai}, \mathbf{x}_{bi})_o\}_{o=1}^{K}$ with the maximum distance $d(a,b) = 1 - \text{sim}(\mathbf{e}_{ai}, \mathbf{e}_{bi})$, where $\text{sim}(\mathbf{e}_{ai}, \mathbf{e}_{bi}) = \mathbf{e}_{ai} \cdot \mathbf{e}_{bi}/(\|\mathbf{e}_{ai}\|\|\mathbf{e}_{bi}\|)$ measures cosine similarity, and $\|\cdot\|$ is the Euclidean norm. These samples are then presented to $\Psi$ with a prompt $\mathbf{p}_{\text{intra}}$ (see Appendix D.2) to analyze the underlying intra-modal interactions and identify the contributing factors.

**Inter-modal Contrastive Exploration:** To highlight the inter-modal interactions, we construct cross-modal contrastive pairs of samples with the maximum misalignment. We quantify this mismatch using a combined score $s(a,b) = (1 - \text{sim}(\mathbf{e}_{ai}, \mathbf{e}_{bj})) + (|y_i - y_j|)$ that considers both semantic distance in subspace and the difference in their scores of the target variable $Y$. We present the top-$K$ pairs $\mathcal{P}_{\text{x}}$ with the highest mismatch scores to the model, along with a prompt $\mathbf{p}_{\text{inter}}$ (see Appendix D.2) that highlights the potential contradiction across modalities. This encourages reasoning about the expected correspondence across modalities and helps pinpoint specific attributes or concepts where this correspondence breaks down, thereby revealing factors governing inter-modal relationships.

**Factor Consolidation and Annotation:** The contrastive analyses generate multiple sets of candidate factors, which may contain redundancies or overlap with previously identified factors $\mathbf{V}^{(t-1)}$. For deduplication, we prompt $\Psi$ with $\mathbf{p}_{\text{m}}$ to merge similar factors and produce a final factor set $\mathbf{V}^{(t)}$. Then, we use the model with prompt $\mathbf{p}_{\text{a}}$ to annotate corresponding values, forming a structured dataset $\mathcal{D}_{\text{S}}^{(t)}$ for causal structure discovery. The overall process of the CFD module is summarized as:

$$\mathbf{V}^{(t)} := \Psi(\Psi(\{\mathcal{P}_i\}_{i=1}^{m}, \mathbf{p}_{\text{intra}}), \Psi(\mathcal{P}_{\text{x}}, \mathbf{p}_{\text{inter}}), \mathbf{V}^{(t-1)}, \mathbf{p}_{\text{m}}), \quad \mathcal{D}_{\text{S}}^{(t)} := \Psi(\mathcal{D}^{(t)}, \mathbf{V}^{(t)}, \mathbf{p}_{\text{a}}). \quad (1)$$

### 3.3 Causal Structure Discovery

With the identified factors $\mathbf{V}^{(t)}$ and their annotated values in $\mathcal{D}_{\text{S}}^{(t)}$, we adopt a statistical CD method $\mathcal{C}$ (e.g., FCI or PC [31]) to infer the causal structure into a causal graph:

$$\mathcal{G}^{(t)} = \mathcal{C}(\mathcal{D}_{\text{S}}^{(t)}, \mathbf{V}^{(t)} \cup \{Y\}). \quad (2)$$

Given the potential presence of unobserved confounders in real-world scenarios, algorithms robust to latent variables, such as FCI algorithm [31] or its variants (e.g., RFCI [79]), are particularly relevant. In this work, we instantiate our framework with the standard FCI algorithm. However, it can be easily adapted with other algorithms [31] for different causal assumptions and data types [80].

While statistical methods provide a principled foundation for structure learning, they rely solely on observational data and thus struggle to address ambiguities [29]. This motivates the need for integrating additional knowledge and reasoning capabilities to refine the discovered structure.

### 3.4 Multimodal Counterfactual Reasoning

The causal graph $\mathcal{G}^{(t)}$ inferred by statistical methods from the derived datasets $\mathcal{D}_{\text{S}}^{(t)}$ often contains ambiguity (e.g., represented by undirected edges in a partial ancestral graph) due to potential limitations of finite and noisy observational data. To alleviate this issue, we introduce the MCR module, which leverages the implicit world knowledge and reasoning capabilities of an MLLM $\Psi$ to refine the causal graph iteratively.

The core idea is to use the MLLM to explore "*what if*" scenarios regarding the uncertainty in $\mathcal{G}^{(t)}$, thereby generating counterfactuals to support genuine causal relationships. By simulating interventions on specific factors and predicting how other factors are affected and how the sample changes, we generate counterfactual samples that implicitly encode causal assumptions derived from MLLM's knowledge. Once validated for plausibility and consistency, they are combined with observational data to provide complementary evidence supporting statistical discovery.

**Counterfactual Generation:** We first identify the uncertain relationships in the inferred graph $\mathcal{G}^{(t)}$. This typically includes factors connected with ambiguous endpoints, indicating uncertainty about the direction of causation or the possibility of latent confounders. These factors then become candidates for counterfactual intervention. For an uncertain factor $V_a$, we prompt $\Psi$ with $\mathbf{p}_{\text{MCR}}$ to perform counterfactual reasoning based on selected samples $\mathcal{P}^{(t)} = \{\mathcal{P}_i^{(t)}\}_{i=1}^{m} \cup \mathcal{P}_{\text{x}}^{(t)}$. The model predicts the counterfactual values $\mathbf{v}_k' = [v_{k1}', \cdots, v_{kd}']$ for $\mathbf{X}_k \in \mathcal{P}^{(t)}$ under the hypothetical scenario

$V_a = v'_{ka}$. Based on the reasoning results, it generates the corresponding multimodal samples $\mathbf{X}'_k$. For instance, it revises the original text $\mathbf{x}_{kt}$ minimally to reflect the changes in factor values, producing a counterfactual text $\mathbf{x}'_{kt}$. For visual counterfactuals, it generates a brief description of the image modifications and leverages an image generation model $\Phi$ (e.g., Gemini 2.0) to produce $\mathbf{x}'_{ki}$. If no changes are needed, then $\mathbf{x}'_{ki} = \mathbf{x}_{ki}$. Formally, this process is summarized as:

$$(\mathbf{X}'_k, \mathbf{v}'_k, y'_k) := \Psi(\mathbf{X}_k, \mathbf{v}_k, V_a = v'_{ka}, \mathbf{p}_{\text{MCR}}; \Phi). \tag{3}$$

Since MLLMs may suffer from hallucination issues [81], we also need to validate the generated counterfactual samples for plausibility and consistency. We mainly apply the following two checks.

**Semantic Plausibility:** We ensure the generated counterfactual sample $\mathbf{S}'_k = (\mathbf{X}'_k, \mathbf{v}'_k, y'_k)$ is semantically coherent and not drastically different from the original sample. This is approximated by measuring the similarity between the original and counterfactual embeddings:

$$I_{\text{sem}}(\mathbf{S}'_k) = \mathbb{I}[\frac{1}{m} \sum_{i=1}^{m} (\text{sim}(\mathbf{e}_{ki}, \mathbf{e}'_{ki})) \geq \tau_{\text{sem}}], \tag{4}$$

where $\mathbb{I}[\cdot]$ is the indicator function, and $\tau_{\text{sem}}$ is a threshold.

**Causal Consistency:** This check verifies if the changes in factor values from $\mathbf{v}_k$ to $\mathbf{v}'_k$ are consistent with the causal structure implied by $\mathcal{G}^{(t)}$. A key principle is that an intervention on $V_a$ should ideally not cause changes in its causal non-descendants, assuming no confounding paths are activated in unexpected ways by the MLLM's reasoning. To assess this, we first identify the set of non-descendants of the intervened variable $V_a$ in $\mathcal{G}^{(t)}$, denoted as $\text{NonDesc}(V_a, \mathcal{G}^{(t)})$. Then, we calculate the proportion of non-descendant nodes that exhibit a significant change. Let $\Delta v_{kj} = |v'_{kj} - v_{kj}|$ for $V_j \in \text{NonDesc}(V_a, \mathcal{G}^{(t)})$, we have

$$R_{\text{indep}}(\mathbf{S}'_k, V_a, \mathcal{G}^{(t)}) = \frac{\Sigma_{V_j \in \text{NonDesc}(V_a, \mathcal{G}^{(t)})} \mathbb{I}[\Delta v_{kj} \geq \epsilon]}{|\text{NonDesc}(V_a, \mathcal{G}^{(t)})|}. \tag{5}$$

The counterfactual $\mathbf{S}'_k$ is considered causally consistent if the proportion of changed non-descendants is below a threshold $\tau_{\text{causal}}$, i.e.,

$$I_{\text{causal}}(\mathbf{S}'_k, V_a, \mathcal{G}^{(t)}) = \mathbb{I}[R_{\text{indep}}(\mathbf{S}'_k, V_a, \mathcal{G}^{(t)}) \leq \tau_{\text{causal}}]. \tag{6}$$

Note that, since the current graph $\mathcal{G}^{(t)}$ used for the check is not ground truth and may contain errors, a non-zero threshold $\tau_{\text{causal}}$ is necessary. This threshold serves as a trade-off between injecting MLLM's knowledge and maintaining consistency with a potentially improvable graph. The validated samples $\mathcal{D}_{\text{CF}}^{(t)}$ with $I_{\text{sem}} \wedge I_{\text{causal}} = 1$ are included for the next round as $\mathcal{D}^{(t+1)} = \mathcal{D}^{(t)} \cup \mathcal{D}_{\text{CF}}^{(t)}$. In this way, the observational dataset is effectively augmented using the MLLM's knowledge and reasoning capabilities, facilitating the iterative refinement of causal factor and structure discovery. The overall process of MLLM-CD is summarized in Algorithm 1 (Appendix C).

Despite the promise of modern MLLMs in demonstrating high-level general knowledge and reasoning capabilities [15], it is important to acknowledge that they may still fall short in challenging scenarios requiring cross-domain or domain-specific expertise. In such cases, we recommend applying stricter validation to filter out noisy counterfactuals and integrating with strategies like retrieval-augmented generation (RAG) [82, 83] or knowledge graph (KG) grounding [84, 85] to enhance the model's knowledge base and reasoning capabilities.

## 4 Experiments

We evaluate both the causal factor and structure discovery performance of MLLM-CD on both synthetic and real-world multimodal datasets, based on state-of-the-art multimodal LLMs including GPT-4o [86], Gemini 2.0 [87], LLaMA 4 Maverick [88], and Grok-2v [89]. Due to the space limit, full comparison results and detailed analyses, including more ablation studies, parameter analysis, time complexity and scalability analysis, and case studies, are provided in Appendix D.

### 4.1 Experimental Setup

We construct two multimodal datasets for evaluation: (1) **Multimodal Apple Gastronome** (MAG) dataset, which is a synthetic dataset with 200 samples with 9 high-level factors, and (2) **Lung Cancer**

Table 1: Causal factor identification and structure discovery performance on the MAG dataset.[2]

| LLM | Method | NP ↑ | NR ↑ | NF ↑ | AP ↑ | AR ↑ | AF ↑ | ESHD ↓ |
|---|---|---|---|---|---|---|---|---|
| GPT-4o | META | $0.45 \pm 0.08$ | $0.52 \pm 0.06$ | $0.48 \pm 0.07$ | **$0.72 \pm 0.05$** | $0.37 \pm 0.06$ | $0.49 \pm 0.04$ | $24.33 \pm 3.51$ |
| | Pairwise | - | - | - | $0.56 \pm 0.10$ | $0.33 \pm 0.11$ | $0.41 \pm 0.11$ | $43.33 \pm 6.43$ |
| | Triplet | - | - | - | $0.33 \pm 0.06$ | $0.37 \pm 0.06$ | $0.35 \pm 0.06$ | $61.67 \pm 5.77$ |
| | COAT | $0.78 \pm 0.19$ | $0.37 \pm 0.13$ | $0.49 \pm 0.15$ | $0.37 \pm 0.32$ | $0.19 \pm 0.17$ | $0.25 \pm 0.22$ | $18.67 \pm 2.52$ |
| | MLLM-CD | **$0.83 \pm 0.11$** | **$0.85 \pm 0.06$** | **$0.84 \pm 0.09$** | $0.69 \pm 0.05$ | **$0.41 \pm 0.06$** | **$0.51 \pm 0.04$** | **$15.33 \pm 2.31$** |
| Gemini 2.0 | META | $0.71 \pm 0.07$ | $0.63 \pm 0.06$ | $0.67 \pm 0.07$ | $0.69 \pm 0.08$ | $0.41 \pm 0.13$ | $0.51 \pm 0.11$ | $18.67 \pm 2.31$ |
| | Pairwise | - | - | - | $0.49 \pm 0.09$ | $0.56 \pm 0.11$ | $0.51 \pm 0.06$ | $30.00 \pm 2.00$ |
| | Triplet | - | - | - | $0.41 \pm 0.02$ | **$0.59 \pm 0.13$** | $0.48 \pm 0.05$ | $32.00 \pm 2.00$ |
| | COAT | $0.85 \pm 0.13$ | $0.41 \pm 0.06$ | $0.51 \pm 0.09$ | $0.69 \pm 0.10$ | $0.26 \pm 0.06$ | $0.37 \pm 0.05$ | $16.00 \pm 1.00$ |
| | MLLM-CD | **$0.86 \pm 0.05$** | **$0.89 \pm 0.00$** | **$0.87 \pm 0.03$** | **$0.76 \pm 0.08$** | $0.52 \pm 0.13$ | **$0.60 \pm 0.06$** | **$14.00 \pm 3.46$** |
| LLaMA 4 | META | $0.51 \pm 0.06$ | $0.41 \pm 0.06$ | $0.45 \pm 0.05$ | $0.81 \pm 0.17$ | $0.26 \pm 0.06$ | $0.39 \pm 0.07$ | $21.67 \pm 0.58$ |
| | Pairwise | - | - | - | $0.50 \pm 0.17$ | $0.30 \pm 0.06$ | $0.36 \pm 0.04$ | $34.67 \pm 6.66$ |
| | Triplet | - | - | - | $0.66 \pm 0.32$ | $0.30 \pm 0.06$ | $0.38 \pm 0.04$ | $34.33 \pm 7.77$ |
| | COAT | **$1.00 \pm 0.00$** | $0.41 \pm 0.06$ | $0.58 \pm 0.07$ | **$0.89 \pm 0.19$** | $0.30 \pm 0.06$ | $0.44 \pm 0.10$ | $14.67 \pm 1.15$ |
| | MLLM-CD | **$1.00 \pm 0.00$** | **$0.85 \pm 0.06$** | **$0.92 \pm 0.04$** | $0.62 \pm 0.08$ | **$0.59 \pm 0.06$** | **$0.60 \pm 0.04$** | **$13.33 \pm 0.58$** |
| Grok-2v | META | $0.56 \pm 0.11$ | $0.44 \pm 0.00$ | $0.49 \pm 0.04$ | $0.75 \pm 0.00$ | $0.33 \pm 0.00$ | $0.46 \pm 0.00$ | $20.33 \pm 3.06$ |
| | Pairwise | - | - | - | $0.35 \pm 0.02$ | $0.33 \pm 0.00$ | $0.34 \pm 0.01$ | $28.33 \pm 11.37$ |
| | Triplet | - | - | - | $0.32 \pm 0.02$ | $0.33 \pm 0.00$ | $0.33 \pm 0.01$ | $30.33 \pm 12.34$ |
| | COAT | **$1.00 \pm 0.00$** | $0.37 \pm 0.17$ | $0.53 \pm 0.18$ | $0.17 \pm 0.29$ | $0.07 \pm 0.13$ | $0.10 \pm 0.18$ | $16.33 \pm 1.15$ |
| | MLLM-CD | **$1.00 \pm 0.00$** | **$0.85 \pm 0.06$** | **$0.92 \pm 0.04$** | **$0.79 \pm 0.21$** | **$0.44 \pm 0.00$** | **$0.56 \pm 0.06$** | **$11.00 \pm 2.65$** |
| Average | META | $0.56 \pm 0.12$ | $0.50 \pm 0.10$ | $0.52 \pm 0.10$ | **$0.74 \pm 0.10$** | $0.34 \pm 0.09$ | $0.46 \pm 0.07$ | $21.25 \pm 3.11$ |
| | Pairwise | - | - | - | $0.47 \pm 0.12$ | $0.38 \pm 0.13$ | $0.40 \pm 0.09$ | $34.08 \pm 8.76$ |
| | Triplet | - | - | - | $0.43 \pm 0.20$ | $0.40 \pm 0.14$ | $0.39 \pm 0.07$ | $39.58 \pm 15.00$ |
| | COAT | $0.91 \pm 0.14$ | $0.39 \pm 0.10$ | $0.53 \pm 0.11$ | $0.53 \pm 0.36$ | $0.20 \pm 0.13$ | $0.29 \pm 0.19$ | $16.42 \pm 2.02$ |
| | MLLM-CD | **$0.92 \pm 0.10$** | **$0.86 \pm 0.05$** | **$0.89 \pm 0.06$** | $0.72 \pm 0.12$ | **$0.49 \pm 0.10$** | **$0.57 \pm 0.06$** | **$13.42 \pm 2.68$** |

dataset, which is a real-world dataset with 60 samples with 5 high-level factors. Please refer to Appendix D.3 for more details on the experimental settings and environment information.

**Multimodal Apple Gastronome:** Following the practice in work [20], we generate a multimodal version of the Apple Gastronome dataset with 200 samples, where 3 visual factors (i.e., `color`, `size`, and `defects`) and 5 verbal factors (i.e., `aroma`, `taste`, `juiciness`, `nutrition`, and `recmd`). Each sample represents an apple with a specific combination of these attributes. The target variable is the overall score of the apple, which is influenced by the other factors as shown in the ground truth causal graph in Figure 3 (a). The visual factors (shown as blue nodes) are represented by images of apples, which are generated by Stable Diffusion 3.5 [90], while the verbal factors (shown as black nodes) are represented by a review text generated by Gemini 2.0 [87].

**Lung Cancer:** We construct a real-world multimodal dataset for lung cancer diagnosis. The samples are collected from the MedPix® database [3]. We select 60 representative lung cancer cases (e.g., Non-Small Cell Lung Cancer [91]) with 5 high-level factors. (1) verbal factors: `gender` and `age` from the demographic data; `smoking` from the history data; and (2) visual factors: `lesion` from the medical images (e.g., CT scans). The target variable is the `diagnosis` of lung cancer, which is affected by other factors in the way of the causal graph shown in Figure 4 (a). More details on the dataset construction are provided in Appendix D.1.

**Baselines:** **META** [20] performs zero-shot factor and structure proposal given only contextual information to LLM; **Pairwise** [29] and **Triplet** [60] use LLM to infer causal relationships among pairwise or triplet variables; **COAT** [20] is the state-of-the-art LLM-driven unstructured CD method and MLLMs are used to adapt it to multimodal settings. The original implementation of COAT only focuses on the discovery of Markov Blanket. We extend it to discovering the entire causal graph for fair comparison. Note that while other LLM-driven CD methods exist, e.g., [41, 48–55, 57, 92–97], they only focus on structural CD and cannot be directly adapted to unstructured multimodal data.

**Metrics:** Following the common practice in CD [30], we evaluate the performance of both factor identification and causal structure discovery with commonly used metrics. For factor identification, we use: (1) **Node Precision** (NP), (2) **Node Recall** (NR), and (3) **Node F1-score** (NF). For causal structure discovery, we adopt (1) **Adjacency Precision** (AP), (2) **Adjacency Recall** (AR), and (3) **Adjacency F1-score** (AF). To jointly evaluate both perspectives, we use an extended Structural

---

[2]Pairwise and Triplet do not involve the step of factor identification, thus metrics related to factor identification are not applicable. Instead, we use the factors discovered by META for their structure discovery.

[3]`https://medpix.nlm.nih.gov/home`

Hamming Distance [98] (**ESHD**), which incorporates additional penalization for mismatched nodes and edges. More details are offered in Appendix D.3.

## 4.2 Analysis with Multimodal Apple Gastronome Dataset

The empirical results of causal discovery on the MAG dataset are shown in Table 1 and Figure 3. The full comparison results are available in Appendix D.4. MLLM-CD overall outperforms baselines in most cases.

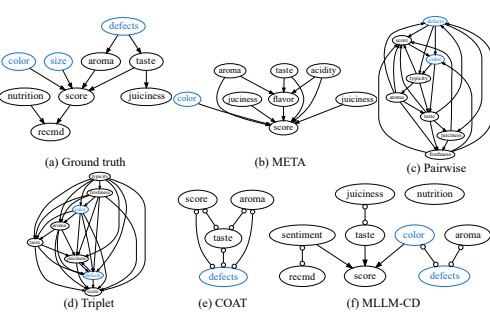

**Factor Identification:** MLLM-CD achieves the most accurate and comprehensive factor identification in terms of quantitative metrics (NP, NR, NF) and qualitative results (Figure 3 (f)). This is attributed to the contrastive factor discovery, which effectively guides the MLLM to identify factors by exploring intra-modal and inter-modal interactions. Meanwhile, we observe that MLLM-CD can

Figure 3: Discovered graphs on MAG dataset.

achieve satisfactory factor identification performance in the first iteration, indicating the efficiency of the CFD module. In contrast, COAT fails to identify sufficient factors and has a lower recall rate.

**Structure Discovery:** MLLM-CD achieves the highest adjacency evaluation scores (AP, AR, AF) in most cases and has the lowest ESHD score, indicating that the discovered causal graph is closer to the ground truth. The improvement is mainly due to the multimodal counterfactual reasoning module, which refines the causal structure by leveraging the MLLM's knowledge and reasoning capabilities. COAT, on the other hand, can hardly determine certain causal relationships. Although META has lower factor identification performance, it successfully identifies several correct and certain causal relationships based on its general knowledge, showing the potential of causal reasoning in MLLMs.

## 4.3 Analysis with Lung Cancer Dataset

Apart from the synthetic dataset, we also evaluate the performance of MLLM-CD on the real-world Lung Cancer dataset. The results are shown in Table 2 and Figure 4. Consistent with the results on the MAG dataset, MLLM-CD achieves the best quantitative performance in most cases, especially in terms of ESHD, showing the overall improvement of MLLM-CD over the baselines. The qualitative results show that MLLM-CD accurately identifies the causal relationships between all the factors and the target variable. COAT achieves comparable performance but includes incorrect directions and noisy/missing factors. Since META,

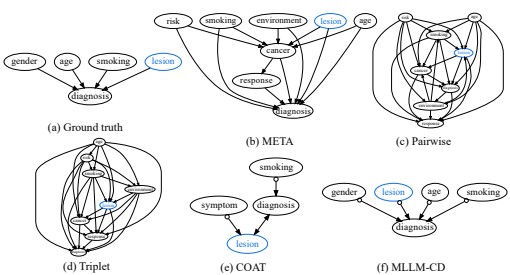

Figure 4: Discovered causal graphs on Lung Cancer dataset.

Pairwise, and Triplet solely rely on MLLM's knowledge, the generated causal graphs are certain but often overly complex with multiple redundant edges. The results indicate that MLLM-CD is capable of effectively identifying the causal factors and structures in real-world multimodal datasets, demonstrating its practical applicability.

## 4.4 Ablation Study

We conduct an ablation study to evaluate the effectiveness of each module in MLLM-CD. We compare the full MLLM-CD with the following variants: (1) **MLLM-CD w/o Contrastive Factor Discovery** (w/o CFD): This variant does not perform contrastive factor discovery, and instead, only uses a simple method, similar to [20], for factor identification. (2) **MLLM-CD w/o Counterfactual Reasoning** (w/o CR): This variant only uses the contrastive factor discovery module and the causal structure discovery module. (3) **MLLM-CD w/o Both** (w/o Both): This variant only uses a simple factor identification prompt from [20] and then performs causal structure discovery without any multimodal reasoning. The performance of these variants with Gemini 2.0 on two datasets is shown in Table 3.

Table 2: Causal factor identification and structure discovery performance on the Lung Cancer dataset.[2]

| LLM | Method | NP ↑ | NR ↑ | NF ↑ | AP ↑ | AR ↑ | AF ↑ | ESHD ↓ |
|---|---|---|---|---|---|---|---|---|
| GPT-4o | META | $0.33 \pm 0.05$ | $0.60 \pm 0.00$ | $0.42 \pm 0.04$ | $\mathbf{1.00 \pm 0.00}$ | $0.50 \pm 0.00$ | $0.67 \pm 0.00$ | $21.67 \pm 4.62$ |
| | Pairwise | - | - | - | $0.67 \pm 0.00$ | $0.50 \pm 0.00$ | $0.57 \pm 0.00$ | $36.33 \pm 10.21$ |
| | Triplet | - | - | - | $0.67 \pm 0.00$ | $0.50 \pm 0.00$ | $0.57 \pm 0.00$ | $47.33 \pm 14.98$ |
| | COAT | $0.47 \pm 0.18$ | $0.40 \pm 0.00$ | $0.42 \pm 0.07$ | $0.33 \pm 0.58$ | $0.08 \pm 0.14$ | $0.13 \pm 0.23$ | $11.67 \pm 3.06$ |
| | MLLM-CD | $\mathbf{0.89 \pm 0.10}$ | $\mathbf{1.00 \pm 0.00}$ | $\mathbf{0.94 \pm 0.05}$ | $0.92 \pm 0.14$ | $\mathbf{0.58 \pm 0.14}$ | $\mathbf{0.69 \pm 0.05}$ | $\mathbf{5.00 \pm 0.00}$ |
| Gemini 2.0 | META | $0.43 \pm 0.07$ | $0.73 \pm 0.12$ | $0.54 \pm 0.08$ | $0.92 \pm 0.14$ | $0.67 \pm 0.14$ | $0.76 \pm 0.10$ | $16.00 \pm 0.00$ |
| | Pairwise | - | - | - | $0.56 \pm 0.10$ | $0.67 \pm 0.14$ | $0.59 \pm 0.02$ | $33.33 \pm 3.51$ |
| | Triplet | - | - | - | $0.56 \pm 0.10$ | $0.67 \pm 0.14$ | $0.59 \pm 0.02$ | $40.00 \pm 8.19$ |
| | COAT | $0.75 \pm 0.25$ | $0.47 \pm 0.12$ | $0.56 \pm 0.11$ | $\mathbf{1.00 \pm 0.00}$ | $0.33 \pm 0.14$ | $0.49 \pm 0.15$ | $8.67 \pm 2.08$ |
| | MLLM-CD | $\mathbf{0.94 \pm 0.10}$ | $\mathbf{1.00 \pm 0.00}$ | $\mathbf{0.97 \pm 0.05}$ | $0.92 \pm 0.14$ | $\mathbf{0.83 \pm 0.14}$ | $\mathbf{0.87 \pm 0.13}$ | $\mathbf{4.67 \pm 0.58}$ |
| LLaMA 4 | META | $0.41 \pm 0.08$ | $0.67 \pm 0.12$ | $0.50 \pm 0.04$ | $0.72 \pm 0.25$ | $0.42 \pm 0.14$ | $0.52 \pm 0.17$ | $19.33 \pm 5.03$ |
| | Pairwise | - | - | - | $0.76 \pm 0.21$ | $0.58 \pm 0.14$ | $0.63 \pm 0.05$ | $32.33 \pm 17.24$ |
| | Triplet | - | - | - | $0.61 \pm 0.10$ | $0.58 \pm 0.14$ | $0.58 \pm 0.02$ | $36.67 \pm 20.40$ |
| | COAT | $0.81 \pm 0.17$ | $0.47 \pm 0.12$ | $0.58 \pm 0.08$ | $\mathbf{1.00 \pm 0.00}$ | $0.25 \pm 0.00$ | $0.40 \pm 0.00$ | $7.67 \pm 0.58$ |
| | MLLM-CD | $\mathbf{0.82 \pm 0.02}$ | $\mathbf{0.93 \pm 0.12}$ | $\mathbf{0.87 \pm 0.06}$ | $0.92 \pm 0.14$ | $\mathbf{0.67 \pm 0.14}$ | $\mathbf{0.76 \pm 0.10}$ | $\mathbf{5.67 \pm 0.58}$ |
| Grok-2v | META | $0.29 \pm 0.04$ | $0.40 \pm 0.00$ | $0.33 \pm 0.03$ | $\mathbf{1.00 \pm 0.00}$ | $\mathbf{0.25 \pm 0.00}$ | $\mathbf{0.40 \pm 0.00}$ | $20.00 \pm 5.29$ |
| | Pairwise | - | - | - | $\mathbf{1.00 \pm 0.00}$ | $\mathbf{0.25 \pm 0.00}$ | $\mathbf{0.40 \pm 0.00}$ | $28.00 \pm 8.89$ |
| | Triplet | - | - | - | $\mathbf{1.00 \pm 0.00}$ | $\mathbf{0.25 \pm 0.00}$ | $\mathbf{0.40 \pm 0.00}$ | $31.00 \pm 8.00$ |
| | COAT | $0.56 \pm 0.21$ | $0.47 \pm 0.23$ | $0.51 \pm 0.22$ | $0.67 \pm 0.58$ | $0.17 \pm 0.14$ | $0.27 \pm 0.23$ | $9.67 \pm 1.53$ |
| | MLLM-CD | $\mathbf{0.87 \pm 0.12}$ | $\mathbf{0.80 \pm 0.00}$ | $\mathbf{0.83 \pm 0.05}$ | $\mathbf{1.00 \pm 0.00}$ | $\mathbf{0.25 \pm 0.00}$ | $\mathbf{0.40 \pm 0.00}$ | $\mathbf{6.00 \pm 1.00}$ |
| Average | META | $0.36 \pm 0.08$ | $0.60 \pm 0.15$ | $0.45 \pm 0.09$ | $0.91 \pm 0.17$ | $0.46 \pm 0.18$ | $0.59 \pm 0.17$ | $19.25 \pm 4.27$ |
| | Pairwise | - | - | - | $0.74 \pm 0.20$ | $0.50 \pm 0.18$ | $0.55 \pm 0.10$ | $32.50 \pm 9.97$ |
| | Triplet | - | - | - | $0.71 \pm 0.19$ | $0.50 \pm 0.18$ | $0.54 \pm 0.08$ | $38.75 \pm 13.36$ |
| | COAT | $0.65 \pm 0.23$ | $0.45 \pm 0.12$ | $0.52 \pm 0.13$ | $0.75 \pm 0.45$ | $0.21 \pm 0.14$ | $0.32 \pm 0.21$ | $9.42 \pm 2.31$ |
| | MLLM-CD | $\mathbf{0.88 \pm 0.09}$ | $\mathbf{0.93 \pm 0.10}$ | $\mathbf{0.90 \pm 0.07}$ | $\mathbf{0.94 \pm 0.11}$ | $\mathbf{0.58 \pm 0.25}$ | $\mathbf{0.68 \pm 0.19}$ | $\mathbf{5.33 \pm 0.78}$ |

Please refer to Appendix D.6 for the full results. One can observe that excluding either the contrastive factor discovery or the counterfactual reasoning module leads to a significant drop in performance. In specific, contrastive factor discovery shows more impact on the accuracy and completeness of factor identification. The counterfactual reasoning module, on the other hand, shows more effectiveness in refining the causal structure, as we can see from the comparison of MLLM-CD w/o CR and the full MLLM-CD.

Table 3: Ablation Study of MLLM-CD on the MAG and Lung Cancer datasets.

| Dataset | Variant | NF ↑ | AF ↑ | ESHD ↓ |
|---|---|---|---|---|
| MAG | w/o CFD | $0.73 \pm 0.07$ | $0.47 \pm 0.09$ | $15.00 \pm 0.00$ |
| | w/o CR | $0.81 \pm 0.09$ | $0.52 \pm 0.06$ | $15.67 \pm 3.51$ |
| | w/o Both | $0.54 \pm 0.08$ | $0.41 \pm 0.08$ | $16.33 \pm 2.08$ |
| | MLLM-CD | $\mathbf{0.87 \pm 0.03}$ | $\mathbf{0.60 \pm 0.06}$ | $\mathbf{14.00 \pm 3.46}$ |
| Lung | w/o CFD | $0.62 \pm 0.04$ | $0.36 \pm 0.34$ | $8.00 \pm 1.00$ |
| | w/o CR | $0.94 \pm 0.05$ | $0.38 \pm 0.04$ | $5.33 \pm 1.53$ |
| | w/o Both | $0.55 \pm 0.11$ | $0.13 \pm 0.23$ | $9.67 \pm 2.31$ |
| | MLLM-CD | $\mathbf{0.97 \pm 0.05}$ | $\mathbf{0.87 \pm 0.13}$ | $\mathbf{4.67 \pm 0.58}$ |

Meanwhile, it contributes to larger improvements on the relatively smaller dataset (Lung Cancer) than the larger one (MAG). This indicates the importance of introducing plausible counterfactuals based on the MLLM's knowledge and reasoning capabilities to alleviate the data scarcity issue and possible data noise. By combining both modules, they jointly contribute to the improvement of causal factor and structure discovery, demonstrating the effectiveness of MLLM-CD.

In addition, we further evaluate the effectiveness of different sampling strategies used in our contrastive factor discovery module. We compare the following variants: (1) **Random Sampling** (Random): This variant randomly samples data points from each category of the target variable. (2) **Simple Pairwise Sampling** (Simple Pair): This variant pairs samples with the largest difference in the target variable. (3) **Intra-modal Contrastive Sampling** (Intra-modal): This variant only considers intra-modal contrastive pairs within each modality. (4) **Inter-modal Contrastive Sampling** (Inter-modal): This variant only considers inter-modal contrastive pairs across differ-

Table 4: Ablation Study of the CFD module on the MAG and Lung Cancer datasets.

| Dataset | Variant | NP ↑ | NR ↑ | NF ↑ |
|---|---|---|---|---|
| MAG | Random | $\mathbf{0.94 \pm 0.10}$ | $0.60 \pm 0.06$ | $0.73 \pm 0.07$ |
| | Simple Pair | $0.79 \pm 0.14$ | $0.70 \pm 0.13$ | $0.75 \pm 0.14$ |
| | Intra-modal | $0.88 \pm 0.13$ | $0.78 \pm 0.11$ | $0.82 \pm 0.12$ |
| | Inter-modal | $0.67 \pm 0.19$ | $0.59 \pm 0.17$ | $0.63 \pm 0.18$ |
| | MLLM-CD | $0.86 \pm 0.05$ | $\mathbf{0.89 \pm 0.00}$ | $\mathbf{0.87 \pm 0.03}$ |
| Lung | Random | $0.65 \pm 0.09$ | $0.60 \pm 0.00$ | $0.62 \pm 0.04$ |
| | Simple Pair | $0.50 \pm 0.00$ | $0.60 \pm 0.00$ | $0.55 \pm 0.00$ |
| | Intra-modal | $0.80 \pm 0.00$ | $0.80 \pm 0.00$ | $0.80 \pm 0.00$ |
| | Inter-modal | $0.53 \pm 0.12$ | $0.53 \pm 0.12$ | $0.53 \pm 0.12$ |
| | MLLM-CD | $\mathbf{0.94 \pm 0.10}$ | $\mathbf{1.00 \pm 0.00}$ | $\mathbf{0.97 \pm 0.05}$ |

ent modalities for sampling. The key quantitative results are shown in Table 4, and we have the following observations: (1) Incorporating finer-grained signals by finding semantic-level contrastive pairs within each modality, the intra-modal contrastive sampling further improves the completeness of factor discovery. However, it often overlooks the cross-modal interactions. (2) Inter-modal contrastive sampling acts as an ideal complement to intra-modal strategy, as it examines across modalities and can effectively identify factors missed by intra-modal. Using inter-modal only has lower quantitative results, because it is prompted to emphasize factors tied to cross-modal interactions, as a complement

to intra-modal strategy, rather than aiming to discover all factors. (3) By combining both intra-modal and inter-modal strategies, MLLM-CD achieves the best overall performance in factor discovery, demonstrating the effectiveness of our contrastive factor discovery module.

## 5 Conclusion

This paper presents MLLM-CD, a novel framework for multimodal causal discovery that leverages the multimodal understanding and reasoning capabilities of MLLMs. MLLM-CD consists of three main components: (1) a contrastive factor discovery module to identify accurate and comprehensive causal factors from multimodal unstructured data in a novel multimodal contrastive manner, (2) a causal structure discovery module to learn the causal relationships among the identified factors, and (3) a multimodal counterfactual reasoning module to refine the discovered results by generating and validating counterfactual samples. We conduct extensive experiments on both synthetic and real-world multimodal datasets, demonstrating the superior performance in causal factor and structure discovery and the practical applicability of MLLM-CD.

**Limitations.** While promising, our approach faces several limitations. A primary challenge is the scarcity of large-scale, diverse benchmark datasets for multimodal causal discovery. While the datasets used in this work are thoughtfully curated, they are constrained by small sample sizes, and their ground truth causal graphs depend on domain expert knowledge. Secondly, the range of modalities MLLM-CD can effectively process is bound by the capabilities of the core MLLM, presenting challenges for integrating specialized modalities like sensor data or genomic sequences. Finally, the overall effectiveness of the proposed framework is dependent on the sophisticated reasoning and generation capabilities of MLLMs, which may still exhibit hallucinations, reflect biases from their training data, or fail to capture subtle causal nuances, despite recent advancements in model alignment.

**Future Work.** These limitations open several avenues for future research. A crucial direction is the development of larger, more diverse, and rigorously annotated benchmark datasets, potentially using semi-automated annotation or simulation-based approaches, which will also allow for exploring the scalability of MLLM-CD. We will also focus on enhancing the framework to handle a wider range of modalities by developing more sophisticated multimodal fusion and cross-modal reasoning mechanisms. To address the reliance on MLLM capabilities, we plan to develop methods to rigorously validate and improve MLLM outputs within the causal discovery pipeline. This includes integrating with interventional analysis, designing uncertainty quantification for generations, and exploring techniques to mitigate biases and hallucinations specifically for causal reasoning.

Please refer to Appendix B for more discussions on limitations and future work.

## Acknowledgments and Disclosure of Funding

We sincerely appreciate the reviewers' valuable feedback. This work is partially supported by Australia ARC LP220100453 and ARC DP240100955. TLL is partially supported by the following Australian Research Council projects: FT220100318, DP220102121, LP220100527, LP220200949.

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

# Appendix of MLLM-CD

## Contents

# A    Table of Notations

| Notation | Description |
|---|---|
| $\mathcal{D}$ | The raw multimodal unstructured dataset. |
| $\mathcal{D}_{\mathrm{S}}$ | The structured multimodal dataset. |
| $\mathcal{D}_{\mathrm{S}}^{(t)}$ | The structured multimodal dataset at the $t$-th iteration. |
| $\mathcal{D}_{\mathrm{CF}}^{(t)}$ | The validated counterfactual samples at the $t$-th iteration. |
| $\mathbf{X}_k$ | The $k$-th multimodal sample in the dataset $\mathcal{D}$. |
| $\mathbf{X}_k'$ | The generated counterfactual multimodal sample of $\mathbf{X}_k$. |
| $\mathbf{x}_{ki}$ | The $i$-th modality of the $k$-th multimodal sample $\mathbf{X}_k$. |
| $\mathbf{x}_{ki}'$ | The generated counterfactual of the $i$-th modality of $\mathbf{X}_k'$. |
| $\mathcal{C}$ | The statistical causal discovery algorithm. |
| $\mathbf{V}^*$ | The set of ground truth factors. |
| $\mathbf{V}$ | The set of discovered factors. |
| $V_j$ | The $j$-th discovered factor. |
| $V_j^*$ | The $j$-th ground truth factor. |
| $v_{kj}$ | The annotated value of the $j$-th factor in the $k$-th multimodal sample. |
| $v_{kj}'$ | The counterfactual value of the $j$-th factor in the $k$-th multimodal sample. |
| $\mathbf{v}_k$ | The factor values of the multimodal sample $\mathbf{X}_k$. |
| $\mathbf{v}_k'$ | The counterfactual factor values of the $k$-th multimodal sample. |
| $\mathbf{V}^{(t)}$ | The set of discovered factors at the $t$-th iteration. |
| $Y$ | The target variable. |
| $y_k$ | The value of the target variable of the $k$-th multimodal sample. |
| $y_k'$ | The counterfactual value of the target variable of the $k$-th multimodal sample. |
| $\mathcal{G}^*$ | The ground truth causal graph. |
| $\mathcal{G}$ | The discovered causal graph. |
| $\mathbf{E}^*$ | The ground truth edges in $\mathcal{G}^*$. |
| $\mathbf{E}$ | The discovered edges in $\mathcal{G}$. |
| $\mathbf{e}_{ki}$ | The semantic representation of $\mathbf{x}_{ki}$. |
| $\mathbf{e}_{ki}'$ | The semantic representation of the generated counterfactual $\mathbf{x}_{ki}'$. |
| $f_i(\cdot)$ | The foundation model used for extracting semantic representations of $i$-th modality. |
| $\Psi$ | The MLLM for contrastive factor discovery and multimodal counterfactual reasoning. |
| $\Phi$ | The image generation model for visual counterfactual generation. |
| $\mathcal{P}_i$ | The set of intra-modal contrastive pairs in the $i$-th modality. |
| $\mathcal{P}_{\mathrm{x}}$ | The set of inter-modal contrastive pairs. |
| $\mathcal{P}^{(t)}$ | The union set of selected intra- and inter-modal contrastive pairs at the $t$-th iteration. |
| $\mathbf{p}_{\mathrm{intra}}$ | The prompt for intra-modal contrastive exploration. |
| $\mathbf{p}_{\mathrm{inter}}$ | The prompt for inter-modal contrastive exploration. |
| $\mathbf{p}_{\mathrm{m}}$ | The prompt for factor consolidation. |
| $\mathbf{p}_{\mathrm{a}}$ | The prompt for factor annotation. |
| $\mathbf{p}_{\mathrm{MCR}}$ | The prompt for multimodal counterfactual reasoning. |
| $\tau_{\mathrm{sem}}$ | The threshold for semantic plausibility check. |
| $\tau_{\mathrm{causal}}$ | The threshold for causal consistency check. |

# B    Limitations and Future Opportunities

While MLLM-CD demonstrates significant promise in revealing causality from multimodal unstructured data, several limitations present avenues for future research and development.

## B.1    Data Availability, Scale and Quality

**Limitation:** Multimodal unstructured causal discovery is still in its early stages, with a limited availability of benchmark datasets. In particular, there is a notable lack of large-scale, diverse, and publicly accessible unstructured multimodal datasets tailored for causal discovery. While the datasets used in this work are thoughtfully curated, they are constrained by small sample sizes, and their ground truth causal graphs depend on domain expert knowledge [91, 99–101], which can be subjective, incomplete, and costly to scale.

**Future Opportunity:** Developing larger, more diverse, and rigorously annotated benchmark datasets for multimodal unstructured causal discovery. This could involve semi-automated annotation methods [102], leveraging simulations with known causal grounds, or creating platforms for community-driven dataset creation and ground truth curation. Additionally, exploring the scalability of MLLM-CD to larger datasets is a crucial direction for future research. Please refer to Section D.9 for a detailed discussion on the time complexity and scalability of MLLM-CD.

## B.2    Modality Coverage

**Limitation:** The range of modalities effectively processed by MLLM-CD is principally guided by the capabilities of the leveraged MLLMs. While current MLLMs support an expanding array of common modalities (e.g., text, image, audio), future advancements in MLLM architectures will be beneficial for seamlessly integrating a wider spectrum of specialized (e.g., sensor data, genomic sequences, tabular data embedded within reports) or highly heterogeneous data types. This will enable MLLM-CD to tackle an even broader range of real-world problems.

**Future Opportunity:** Future work could explore techniques for more sophisticated multimodal fusion and cross-modal reasoning, facilitating the discovery of causal relations spanning more diverse modalities.

## B.3    MLLM Capabilities and Assumptions

**Limitation:** As we discussed in the theoretical analysis, the effectiveness of MLLM-CD is affected by the sophisticated reasoning and generation capabilities inherent in MLLMs. These could include faithful interpretation, consistent logical reasoning, accurate generation of causally plausible counterfactuals across modalities, and robust understanding of intra- and inter-modal interactions. Despite the rapid advancements in alignment research [103, 104], MLLMs may still hallucinate, exhibit biases present in their training data, or fail to capture subtle causal nuances.

**Future Opportunity:** Developing methods to rigorously validate, calibrate, and enhance MLLM outputs within the causal discovery pipeline. This includes designing more robust prompting strategies, incorporating uncertainty quantification for MLLM-derived factors and relationships, and exploring techniques to mitigate MLLM biases and hallucinations specifically in the context of causal reasoning. Investigating how to formally verify if MLLM reasoning aligns with established causal principles is a crucial area.

# C  Algorithm

---

**Algorithm 1** The MLLM-CD Framework

---

**Require:** Multimodal unstructured dataset $\mathcal{D}$; MLLM $\Psi$; Image generation model $\Phi$; Statistical CD method $\mathcal{C}$; Target $Y$; Maximum number of iterations $T$; Prompts $\mathbf{p}_{\text{intra}}, \mathbf{p}_{\text{inter}}, \mathbf{p}_{\text{m}}, \mathbf{p}_{\text{a}}, \mathbf{p}_{\text{MCR}}$; Thresholds $\tau_{\text{sem}}, \tau_{\text{causal}}, \epsilon$; Number of contrastive pairs $K$.

**Ensure:** Final discovered causal factors $\mathbf{V}^{(T)}$; Final discovered causal graph $\mathcal{G}^{(T)}$.

1: Initialization: $t \leftarrow 1, \mathcal{D}^{(1)} \leftarrow \mathcal{D}, \mathbf{V}^{(0)} \leftarrow \emptyset$.
2: **while** not converged and $t \leq T$ **do**
3:     **// Contrastive Factor Discovery**
4:     Extract semantic representations $\{\mathbf{e}_{ki}\}$ for all samples $\mathbf{X}_k \in \mathcal{D}^{(t)}$ using $f_i(\cdot)$.
5:     For each modality $i = 1, \cdots, m$:
6:         Select top-$K$ intra-modal contrastive pairs $\mathcal{P}_i^{(t)} = \{(\mathbf{x}_{ai}, \mathbf{x}_{bi})_o\}_{o=1}^K$ from $\mathcal{D}^{(t)}$.
7:     Candidate factors from intra-modal contrastive exploration $\mathbf{V}_{\text{intra}}^{(t)} \leftarrow \Psi(\{\mathcal{P}_i^{(t)}\}_{i=1}^m, \mathbf{p}_{\text{intra}})$.
8:     Select top-$K$ cross-modal contrastive pairs $\mathcal{P}_{\text{x}}^{(t)}$ from $\mathcal{D}^{(t)}$.
9:     Candidate factors from inter-modal contrastive exploration $\mathbf{V}_{\text{inter}}^{(t)} \leftarrow \Psi(\mathcal{P}_{\text{x}}^{(t)}, \mathbf{p}_{\text{inter}})$.
10:    Consolidate factors: $\mathbf{V}^{(t)} \leftarrow \Psi(\mathbf{V}_{\text{intra}}^{(t)}, \mathbf{V}_{\text{inter}}^{(t)}, \mathbf{V}^{(t-1)}, \mathbf{p}_{\text{m}})$.
11:    Annotate values: $\mathcal{D}_{\text{S}}^{(t)} \leftarrow \Psi(\mathcal{D}^{(t)}, \mathbf{V}^{(t)}, \mathbf{p}_{\text{a}})$.
12:    **// Causal Structure Discovery**
13:    Infer causal graph: $\mathcal{G}^{(t)} \leftarrow \mathcal{C}(\mathcal{D}_{\text{S}}^{(t)}, \mathbf{V}^{(t)} \cup \{Y\})$.
14:    **// Multimodal Counterfactual Reasoning**
15:    Initialize set of validated counterfactuals $\mathcal{D}_{\text{CF}}^{(t)} \leftarrow \emptyset$.
16:    Identify candidate factors connected with ambiguous endpoints in $\mathcal{G}^{(t)}$.
17:    Let the set of candidate samples for counterfactual reasoning be $\mathcal{P}^{(t)} \leftarrow \{\mathcal{P}_i^{(t)}\}_{i=1}^m \cup \mathcal{P}_{\text{x}}^{(t)}$.
18:    **for** each sample $\mathbf{X}_k$ in $\mathcal{P}^{(t)}$ **do**
19:        **for** each uncertain factor $V_a$ **do**
20:            Generate counterfactual: $(\mathbf{X}_k', \mathbf{v}_k', y_k') \leftarrow \Psi(\mathbf{X}_k, \mathbf{v}_k, y_k, V_a = v_{ka}', \mathbf{p}_{\text{MCR}}; \Phi)$.
21:            Extract embeddings $\mathbf{e}_{ki}'$ for $\mathbf{X}_k'$.
22:            $I_{\text{sem}}(\mathbf{S}_k') \leftarrow \mathbb{I}\left[\frac{1}{m}\sum_{j=1}^m (\text{sim}(\mathbf{e}_{kj}, \mathbf{e}_{kj}')) \geq \tau_{\text{sem}}\right]$.
23:            $R_{\text{indep}}(\mathbf{S}_k', V_a, \mathcal{G}^{(t)}) \leftarrow \frac{\Sigma_{V_j \in \text{NonDesc}(V_a, \mathcal{G}^{(t)})} \mathbb{I}[\Delta v_{kj} \geq \epsilon]}{|\text{NonDesc}(V_a, \mathcal{G}^{(t)})|}$.
24:            $I_{\text{causal}}(\mathbf{S}_k', V_a, \mathcal{G}^{(t)}) \leftarrow \mathbb{I}[R_{\text{indep}}(\mathbf{S}_k', V_a, \mathcal{G}^{(t)}) \leq \tau_{\text{causal}}]$.
25:            **if** $I_{\text{sem}}(\mathbf{S}_k') \wedge I_{\text{causal}}(\mathbf{S}_k', V_a) = 1$ **then**
26:                $\mathcal{D}_{\text{CF}}^{(t)} \leftarrow \mathcal{D}_{\text{CF}}^{(t)} \cup \{\mathbf{X}_k'\}$.     ▷ Store multimodal counterfactual samples
27:            **end if**
28:        **end for**
29:    **end for**
30:    $\mathcal{D}^{(t+1)} \leftarrow \mathcal{D}^{(t)} \cup \mathcal{D}_{\text{CF}}^{(t)}$.     ▷ Augment multimodal dataset
31:    $t \leftarrow t + 1$.
32: **end while**
33: **return** $\mathbf{V}^{(T)}, \mathcal{G}^{(T)}$.

---

# D   More Details about Experiments

## D.1   Details on Dataset Construction

**Multimodal Apple Gastronome (MAG):** Following [20], we construct a multimodal version of the Apple Gastronome dataset [20]. We consider several key factors from different modalities to contribute to the target variable, i.e., the overall rating score of the apple by gastronomes. These factors include 3 visual features (i.e., `color`, `size`, and `defects`) and 5 verbal features (i.e., `aroma`, `taste`, `juiciness`, `nutrition`, and `recmd`). The ground truth definitions of these factors are as follows:

- `color`: The color of the apple, which can be bright red (positive), or greenish (negative).

- `size`: The size of the apple, which can be large (positive), or small (negative).

- `defects`: The presence of defects on the apple's surface, which can be "free of defects" (positive), or "with noticeable defects" (negative).

- `aroma`: The aroma of the apple, which can be strong (positive), or musty/rotten (negative).

- `taste`: The taste of the apple, which can be sweet (positive), or sour (negative).

- `juiciness`: The juiciness of the apple, which can be "abundant and refreshing moisture" (positive), or "dry and lacking moisture" (negative).'

- `nutrition`: The nutritional value of the apple, which can be "highly nutritious with essential nutrients" (positive), or "relatively low in nutritional value" (negative).

- `recmd`: The market potential of the apple, which can be "has significant market potential and deserves wider recognition" (positive), or "might not bring the expected returns and could even lead to losses" (negative).

We use the Apple Gastronome construction script from the COAT project [4] to randomly generate 200 samples, each representing a different combination of factor values. These samples constitute the ground truth structured data and are generated according to the causal relationships defined in Figure 5 (a). We also show the faithful causal graph discovered by the FCI algorithm [31] in Figure 5 (b), inferred directly from the ground truth structured data. Then, we use the Gemini 2.0 model [87] to generate the review text for each sample to present verbal features. The image modality is generated by Stable Diffusion 3.5 [90] to present visual features. The generated review text and images are combined to form the final multimodal unstructured dataset.

Examples of the MAG dataset are given in Figure 6. The prompts used for generating review texts and images are shown in Prompts D.1 and D.2. This dataset will be open-sourced under **CC-BY 4.0**.

**Lung Cancer:** This is a real-world dataset collected from the MedPix® database [5] under **Open Database License**. We select 60 representative lung cancer cases (e.g., Non-Small Cell Lung Cancer [91]). Each case involves four factors, i.e., `gender`, `age`, `smoking`, and `lesion` (image modality), which jointly contribute to the likelihood of lung cancer diagnosis, i.e., the target variable `diagnosis`. The definitions of these factors are as follows:

- `gender`: The patient's gender, which is extracted from the demographic information and can be male (positive) or female (negative). Gender differences in lung cancer incidence have been observed [99, 100], with males historically showing higher rates, lower survival rates at one and five years, and significantly increased risk of mortality.

- `age`: The patient's age, which is extracted from the demographic information and can be "$\geq 60$" (positive) or "$< 60$" (negative). Lung cancer incidence increases with age, with most cases occurring in individuals aged 60 or older [100, 101].

- `smoking`: The patient's smoking history, which is extracted from the medical history and can be "smoker" (positive) or "non-smoker" (negative). Smoking remains the main risk factor for lung cancer [91].

---

[4] https://causalcoat.github.io/
[5] https://medpix.nlm.nih.gov/home

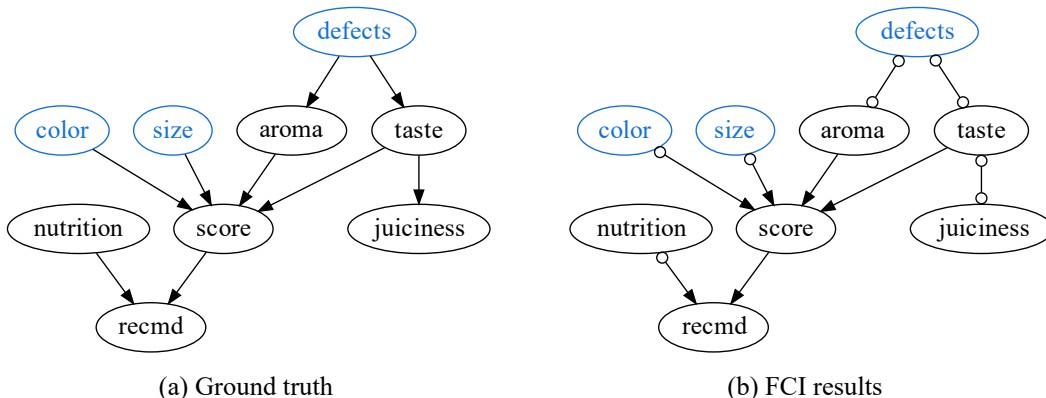

Figure 5: Ground truth and faithful (via FCI algorithm) causal graphs in the MAG dataset.

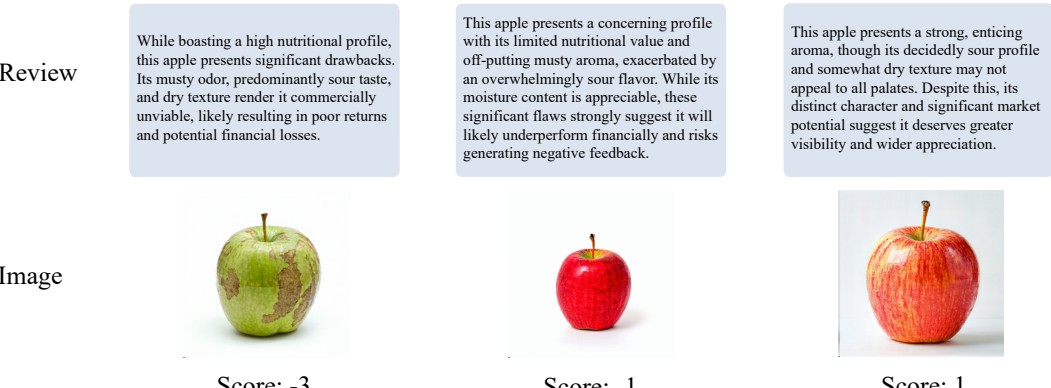

Figure 6: Examples of the MAG dataset.

- `lesion`: The presence of a lesion in the lung, which is extracted from the medical imaging (e.g., CT scans) and can be "with lesion" (positive) or "no clear lesion" (negative). Lesions often present the early visible signs of lung cancer, and imaging techniques such as X-ray, CT, and PET scans are among the most widely used tools for its detection and diagnosis [91].

These factors contribute to the target variable `diagnosis` in the way as shown in the ground truth causal graph in Figure 7 (a). The faithful causal graph discovered by the FCI algorithm [31] is shown in Figure 7 (b). These factors and the causal relationships are carefully curated based on evidence from established medical literature [91, 99–101].

Examples of the Lung Cancer dataset are given in Figure 8.

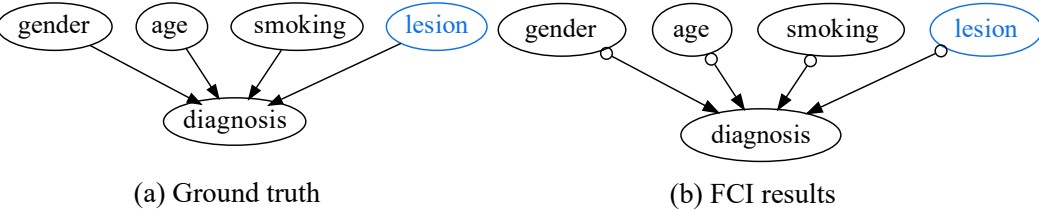

Figure 7: Ground truth and faithful (via FCI algorithm) causal graphs in the Lung Cancer dataset.

|  |  |  |  |
|---|---|---|---|
| Patient Info | 48 year old male with recent history of headache and unsteady gait. | 47 y/o black male with 30 year smoking history, complains of dyspnea on exertion. | An 83-year-old male smoker, COPD on 2L home oxygen, presents to the emergency department with worsening dyspnea on exertion. He had a mild dry cough with an 8 lb weight loss over the past month with no change in appetite. He reported no history of fevers or chills. |

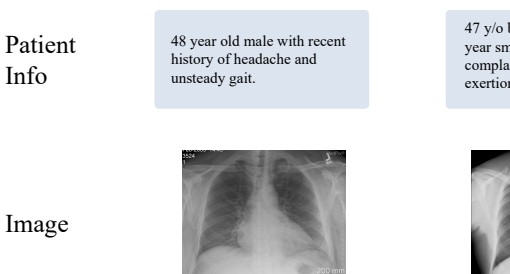
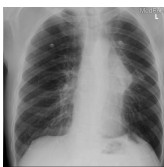

| Image | Diagnosis: 2 | Diagnosis: 3 | Diagnosis: 5 |
|---|---|---|---|

Figure 8: Examples of the Lung Cancer dataset.

## D.2 Details on Prompts

In this section, we provide examples of prompts used in our experiments, including:

- Generating review texts for the MAG dataset (Prompt D.1)

- Generating images for the MAG dataset (Prompt D.2)

- Intra-modal contrastive exploration (Prompt D.3)

- Inter-modal contrastive exploration (Prompt D.4)

- Factor consolidation (Prompt D.5)

- Annotation (Prompt D.6)

- Counterfactual generation (Prompt D.7)

---

**Prompt D.1: Generating Review Texts for MAG Dataset**

You are a picky gastronome on apples. You are ready to **evaluate apples and write reviews**. Your writing should be **clear, solid and convincing to suppliers and customers**.

## Task

Please write a short review about the evaluation results for a given apple.

Evaluation Results: {*apple feature*}

Requirement:
- Combine all of those evaluation results into a more detailed review comment.
- Single paragraph; No quotation marks; The review comments should be complete.
- Modern English.
- No more than 60 words.
- Only output the review comment content directly without any other format or content.

---

**Prompt D.2: Generating Images with Stable Diffusion for MAG Dataset**

A {*size*} apple, that is {*color*}, and {*defects*}, white background, centred, fully in frame, well-lit, realistic photo.

*Negative Prompt:* cropped, partial view, out of frame, low quality, low resolution.

---

## Prompt D.3: An Example of Intra-modal Contrastive Exploration Prompt $p_{intra}$

# Intra-modal Contrastive Analysis

I'm showing you pairs of samples with significant differences.

## Pair 1
### Sample A (ID: {*sample A ID*}):
- **Score**: {*score A*}
- **Review**: {*review A*}
### Sample B (ID: {*sample B ID*}):
- **Score**: {*score B*}
- **Review**: {*review B*}
...

Based on these contrastive pairs, analyze the underlying interactions among potential factors that lead to the observed differences.

# Task: Factor Identification

You are an expert food analyst specializing in apple evaluation. Based on the contrasting pairs and interaction analysis:
1. Identify the key factors that differentiate these samples
2. Focus on factors that can be clearly observed. Each factor should focus on one concrete aspect without overlap with {*factors identified in previous iterations*}
3. Create factors that have clear positive, neutral, and negative criteria

# Output Format

**Part 1**: Explain your analysis about the key differences between each sample pair.
**Part 2**: List the discovered factors using this exact template:

**Factor Name**
- 1: [Positive Criterion]
- 0: [Otherwise; or not mentioned]
- -1: [Negative Criterion]

**Prompt D.4: An Example of Inter-modal Contrastive Exploration Prompt $p_x$**

# Inter-modal Mismatch Analysis

I'm showing you samples where the textual description (review) and visual appearance (image) of samples seem to contradict each other.

## Pair 1
### Text from Sample A (ID: {*sample A ID*}):
- **Score**: {*score A*}
- **Review**: {*review A*}
### Image from Sample B (ID: {*sample B ID*}):
- **Score**: {*score B*}
- **Image**: {*image B*}
...

Notice the potential mismatch between what the review describes and what the image shows. Analyze the underlying interactions among potential factors that could lead to this mismatch.

# Task: Identify Factors that Explain Text-Image Discrepancies

You are an expert food analyst specializing in apple evaluation. Based on the contrastive pairs and interaction analysis:
1. Identify key factors where the textual reviews contradict what's visible in the paired images
2. Each factor should focus on one concrete aspect without overlap{*factors identified in previous iterations*}
3. Create factors that can explain these discrepancies with clear positive, neutral, and negative criteria

# Output Format

**Part 1**: Explain your observations about the mismatches of each pair of textual and visual information.
**Part 2**: List factors that explain these discrepancies using this exact template:

**Factor Name**
- 1: [Positive Criterion]
- 0: [Otherwise; or not mentioned]
- -1: [Negative Criterion]

**Prompt D.5: An Example of Factor Consolidation Prompt $p_m$**

# Factor Deduplication and Refinement

Below is a list of factors identified from different contrastive analyses:

Factor 1: {*factor 1*}
- 1: [Positive Criterion]
- 0: [Neutral Criterion]
- -1: [Negative Criterion]
...

Please:
1. Identify and merge similar factors; Each factor should be distinct and not overlap with others and be specific to one aspect; Avoid general factors like "overall quality"
2. Refine factor definitions for clarity and precision
3. Output a consolidated list of the most important and non-overlapping factors

Only output the final results using this exact format for each factor:

**Factor Name**
- 1: [Positive Criterion]
- 0: [Otherwise; or not mentioned]
- -1: [Negative Criterion]



**Prompt D.6: An Example of Annotation Prompt $p_a$**

# Factor Annotation

Please annotate the following samples based on these factors:

## Factor 1: {*factor 1*}
- 1: [Positive Criterion]
- 0: [Neutral Criterion]
- -1: [Negative Criterion]
...

# Samples to Annotate

## Sample 1:
- **Score**: {*score 1*}
- **Review**: {*review 1*}
- **Image**: {*image 1*}
...

# Task: For each sample, assign a value (-1, 0, or 1) to each factor based on the criteria above.

# Output Format

Please format your response strictly as follows:

**Sample [ID]**:
- Factor 1 ([factor1 name]): [Value]
- Factor 2 ([factor2 name]): [Value]
...

Repeat for all samples.

**Prompt D.7: An Example of Counterfactual Generation Prompt** $p_{MCR}$

# Refining Causal Factors and Relationships

I'm analyzing factors that affect sample scores based on reviews and images. I need your help to refine my understanding of causal relationships.

## Data

### Sample 1:
- **Score**: {*score 1*}
- **Review**: {*review 1*}
- **Image**: {*image 1*}
...

## Current Factors
{*Factor information*}

## Annotated Factors
{*Annotated factor values*}

## Uncertain Causal Relationships
I've identified these causal factors, but there is uncertainty in the causal relationships of the following factors:
{*Uncertain factor information*}

# Task: Counterfactual Reasoning

For each uncertain relationship above, please create a counterfactual scenario: "If factor X were different (i.e., value being reversed if the factor is mentioned and skip the sample if the factor is not mentioned for this sample), how would other factors be affected?" Based on this assumption and your knowledge of apples, predict the values of other factors. Only create counterfactual scenarios that support the valid and reasonable causal relationships and directions. Specifically, there are two types of factors, verbal and visual factors. If the verbal factor is modified, revise the review text directly with minimum changes to reflect the new scenario. If the visual factor is modified, state a short instruction of the changes that need to be applied to the image, e.g., "Change the color of the apple to red." If no visual changes, please state 'N/A'.

# Output Format

Please structure your response in the following format and respectively list the values of all the factors for each sample in each counterfactual scenario:

## Counterfactual Scenario 1: [Changed Factor Name]
**Sample 1**:
- Factor 1: [Value]
- Factor 2: [Value]
...
- Factor N: [Value]
- Review: [Modified Review Text]
- Image: [Image Modification Description] (If applicable, otherwise 'N/A')

**Sample 2**:
...

## Counterfactual Scenario 2: [Changed Factor Name]
...

### D.3 Experimental Settings and Environment

#### D.3.1 Implementation Details

We detail the implementation of MLLM-CD as follows.

In the contrastive factor discovery module, we use the pretrained CLIP model [75] with the ViT-B/32 checkpoint from OpenAI's official release to extract textual and visual embeddings from the multimodal samples in the MAG and Lung Cancer datasets. For intra- and inter-modal contrastive exploration, we choose the top $K = 5$ pairs of samples with the prompts in Section D.2 for factor identification and annotation.

In the causal structure discovery module, we adopt the FCI algorithm [31] to infer the causal structure from the annotated factors. Additional discussion on different CD methods can be found in Section D.7. We use the FCI implementation from the causal-learn library [12], available at the website [6].

In the multimodal counterfactual reasoning module, we set the threshold parameters as $\tau_{\text{sem}} = 0.7$ and $\tau_{\text{causal}} = 0.4$ for consistency validation. $\epsilon$ is a small constant set to $10^{-6}$ to highlight any changes in the non-descendant nodes. Following [20], the maximum number of iterations is set to $T = 3$. Further analysis of parameter choices is discussed in Section D.8. The MLLMs used in experiments are accessed via API calls, and all experiments are conducted on a server with two Intel Xeon 6346 CPUs, 256GB RAM, and two NVIDIA A40 GPUs.

#### D.3.2 Baselines

We provide a detailed list of the representative baselines used in our experiments, including:

- **META** is a strategy from [20] that identifies both causal factors and causal relationships using only the knowledge encoded in MLLMs, guided by contextual information and task descriptions.

- **Pairwise** [29] leverages the knowledge of MLLMs to identify potential causal relationships between the given pairs of factors. Note that, this method only focuses on predicting causal relationships based on given factors, and cannot directly uncover causal factors from unstructured data. In our experiments, we use the factors identified by META as input to this method.

- **Triplet** [60] extends the Pairwise method by using triplet-based queries and a robust aggregation strategy to improve causal relation identification. Similar to Pairwise, it does not involve causal factor discovery, and we use the factors discovered by META as input.

- **COAT** [20] is the most closely related work to ours and, to the best of our knowledge, the only method specifically designed for causal discovery from unstructured data. We use the official implementation of COAT [1] and adapt it to discover the full causal graph using FCI algorithm [31]. Following the original setup [20], we set the number of iterations to $T = 3$.

#### D.3.3 Evaluation Metrics

Two major steps are involved in causal discovery from multimodal unstructured data, i.e., factor identification and structure discovery. To validate the effectiveness of both steps, we adopt the following widely used metrics [30]:

1) **Factor Identification**: We assess the accuracy and comprehensiveness of identified causal factors using precision (**NP**), recall (**NR**), and $F_1$ score (**NF**). Note that, the identified factor names may vary across different methods and models. For evaluation, we manually align semantically correct factors with the ground truth names to ensure consistency.

2) **Structure Discovery**: The quality of the discovered causal structure is evaluated using adjacency precision (**AP**), adjacency recall (**AR**), and their $F_1$ score (**AF**).

3) **Combination**: Different from conventional CD settings where all methods operate on the same predefined set of factors, methods designed for unstructured data may identify different sets of factors, leading to subgraphs that differ in coverage of the ground truth. For a fair and comprehensive

---

[6]https://causal-learn.readthedocs.io/en/latest/index.html

Table 6: The full results for factor identification and structure discovery on the MAG dataset.[7]

| MLLM | Method | NP ↑ | NR ↑ | NF ↑ | AP ↑ | AR ↑ | AF ↑ | ESHD ↓ |
|---|---|---|---|---|---|---|---|---|
| GPT-4o | META | $0.45 \pm 0.08$ | $0.52 \pm 0.06$ | $0.48 \pm 0.07$ | $\mathbf{0.72 \pm 0.05}$ | $0.37 \pm 0.06$ | $0.49 \pm 0.04$ | $24.33 \pm 3.51$ |
| | Pairwise | - | - | - | $0.56 \pm 0.10$ | $0.33 \pm 0.11$ | $0.41 \pm 0.11$ | $43.33 \pm 6.43$ |
| | Triplet | - | - | - | $0.33 \pm 0.06$ | $0.37 \pm 0.06$ | $0.35 \pm 0.06$ | $61.67 \pm 5.77$ |
| | COAT | $0.78 \pm 0.19$ | $0.37 \pm 0.13$ | $0.49 \pm 0.15$ | $0.37 \pm 0.32$ | $0.19 \pm 0.17$ | $0.25 \pm 0.22$ | $18.67 \pm 2.52$ |
| | MLLM-CD | $\mathbf{0.83 \pm 0.11}$ | $\mathbf{0.85 \pm 0.06}$ | $\mathbf{0.84 \pm 0.09}$ | $0.69 \pm 0.05$ | $\mathbf{0.41 \pm 0.06}$ | $\mathbf{0.51 \pm 0.04}$ | $\mathbf{15.33 \pm 2.31}$ |
| Gemini 2.0 | META | $0.71 \pm 0.07$ | $0.63 \pm 0.06$ | $0.67 \pm 0.07$ | $0.69 \pm 0.08$ | $0.41 \pm 0.13$ | $0.51 \pm 0.11$ | $18.67 \pm 2.31$ |
| | Pairwise | - | - | - | $0.49 \pm 0.09$ | $0.56 \pm 0.11$ | $0.51 \pm 0.06$ | $30.00 \pm 2.00$ |
| | Triplet | - | - | - | $0.41 \pm 0.02$ | $\mathbf{0.59 \pm 0.13}$ | $0.48 \pm 0.05$ | $32.00 \pm 2.00$ |
| | COAT | $0.85 \pm 0.13$ | $0.41 \pm 0.06$ | $0.51 \pm 0.09$ | $0.69 \pm 0.10$ | $0.26 \pm 0.06$ | $0.37 \pm 0.05$ | $16.00 \pm 1.00$ |
| | MLLM-CD | $\mathbf{0.86 \pm 0.05}$ | $\mathbf{0.89 \pm 0.00}$ | $\mathbf{0.87 \pm 0.03}$ | $\mathbf{0.76 \pm 0.08}$ | $0.52 \pm 0.13$ | $\mathbf{0.60 \pm 0.06}$ | $\mathbf{14.00 \pm 3.46}$ |
| LLaMA 4 | META | $0.51 \pm 0.06$ | $0.41 \pm 0.06$ | $0.45 \pm 0.05$ | $0.81 \pm 0.17$ | $0.26 \pm 0.06$ | $0.39 \pm 0.07$ | $21.67 \pm 0.58$ |
| | Pairwise | - | - | - | $0.50 \pm 0.17$ | $0.30 \pm 0.06$ | $0.36 \pm 0.04$ | $34.67 \pm 6.66$ |
| | Triplet | - | - | - | $0.66 \pm 0.32$ | $0.30 \pm 0.06$ | $0.38 \pm 0.04$ | $34.33 \pm 7.77$ |
| | COAT | $\mathbf{1.00 \pm 0.00}$ | $0.41 \pm 0.06$ | $0.58 \pm 0.07$ | $\mathbf{0.89 \pm 0.19}$ | $0.30 \pm 0.06$ | $0.44 \pm 0.10$ | $14.67 \pm 1.15$ |
| | MLLM-CD | $\mathbf{1.00 \pm 0.00}$ | $\mathbf{0.85 \pm 0.06}$ | $\mathbf{0.92 \pm 0.04}$ | $0.62 \pm 0.08$ | $\mathbf{0.59 \pm 0.06}$ | $\mathbf{0.60 \pm 0.04}$ | $\mathbf{13.33 \pm 0.58}$ |
| Grok-2v | META | $0.56 \pm 0.11$ | $0.44 \pm 0.00$ | $0.49 \pm 0.04$ | $0.75 \pm 0.00$ | $0.33 \pm 0.00$ | $0.46 \pm 0.00$ | $20.33 \pm 3.06$ |
| | Pairwise | - | - | - | $0.35 \pm 0.02$ | $0.33 \pm 0.00$ | $0.34 \pm 0.01$ | $28.33 \pm 11.37$ |
| | Triplet | - | - | - | $0.32 \pm 0.02$ | $0.33 \pm 0.00$ | $0.33 \pm 0.01$ | $30.33 \pm 12.34$ |
| | COAT | $\mathbf{1.00 \pm 0.00}$ | $0.37 \pm 0.17$ | $0.53 \pm 0.18$ | $0.17 \pm 0.29$ | $0.07 \pm 0.13$ | $0.10 \pm 0.18$ | $16.33 \pm 1.15$ |
| | MLLM-CD | $\mathbf{1.00 \pm 0.00}$ | $\mathbf{0.85 \pm 0.06}$ | $\mathbf{0.92 \pm 0.04}$ | $\mathbf{0.79 \pm 0.21}$ | $\mathbf{0.44 \pm 0.00}$ | $\mathbf{0.56 \pm 0.06}$ | $\mathbf{11.00 \pm 2.65}$ |
| Average | META | $0.56 \pm 0.12$ | $0.50 \pm 0.10$ | $0.52 \pm 0.10$ | $\mathbf{0.74 \pm 0.10}$ | $0.34 \pm 0.09$ | $0.46 \pm 0.07$ | $21.25 \pm 3.11$ |
| | Pairwise | - | - | - | $0.47 \pm 0.12$ | $0.38 \pm 0.13$ | $0.40 \pm 0.09$ | $34.08 \pm 8.76$ |
| | Triplet | - | - | - | $0.43 \pm 0.20$ | $0.40 \pm 0.14$ | $0.39 \pm 0.07$ | $39.58 \pm 15.00$ |
| | COAT | $0.91 \pm 0.14$ | $0.39 \pm 0.10$ | $0.53 \pm 0.11$ | $0.53 \pm 0.36$ | $0.20 \pm 0.13$ | $0.29 \pm 0.19$ | $16.42 \pm 2.02$ |
| | MLLM-CD | $\mathbf{0.92 \pm 0.10}$ | $\mathbf{0.86 \pm 0.05}$ | $\mathbf{0.89 \pm 0.06}$ | $0.72 \pm 0.12$ | $\mathbf{0.49 \pm 0.10}$ | $\mathbf{0.57 \pm 0.06}$ | $\mathbf{13.42 \pm 2.68}$ |

evaluation, we adapt the standard SHD metric [98] to our scenarios, and present the extended SHD (**ESHD**) metric to evaluate the overall performance of causal factor and structure discovery. Apart from the computation of standard SHD on the matched subgraph, it also accounts for missing and spurious factors, as well as the corresponding missing and spurious edges associated with those factors. A lower ESHD value indicates that the discovered graph more closely matches the ground truth, thereby reflecting better overall performance in both factor and structure discovery.

### D.4 The Full Results on MAG Dataset

The full quantitative results for factor identification and structure discovery on the MAG dataset are shown in Table 6. Here, we also illustrate the visual comparison of causal graphs discovered by different methods and MLLM backbones on the MAG dataset. The results are shown in Figures 9, 10, 11 and 12. Several key observations can be made:

1. META, Pairwise, and Triplet infer causal relationships directly from the inherent knowledge of MLLMs. While this leads to non-ambiguity, the resulting graphs tend to be overly complex, often containing numerous redundant edges (e.g., Figure 10 (b)-(d)).

2. COAT identifies only a limited subset of causal factors and relationships. Moreover, the inferred relationships often remain uncertain.

3. MLLM-CD produces a more comprehensive set of causal factors and relationships. Compared to COAT, the structural ambiguities are reduced, and the resulting graph aligns more closely with the ground-truth causal graph and the faithful graph shown in Figure 5 (b).

---

[7]Pairwise and Triplet do not involve the step of factor identification, thus metrics related to factor identification are not applicable. Instead, we use the factors discovered by META for their structure discovery.

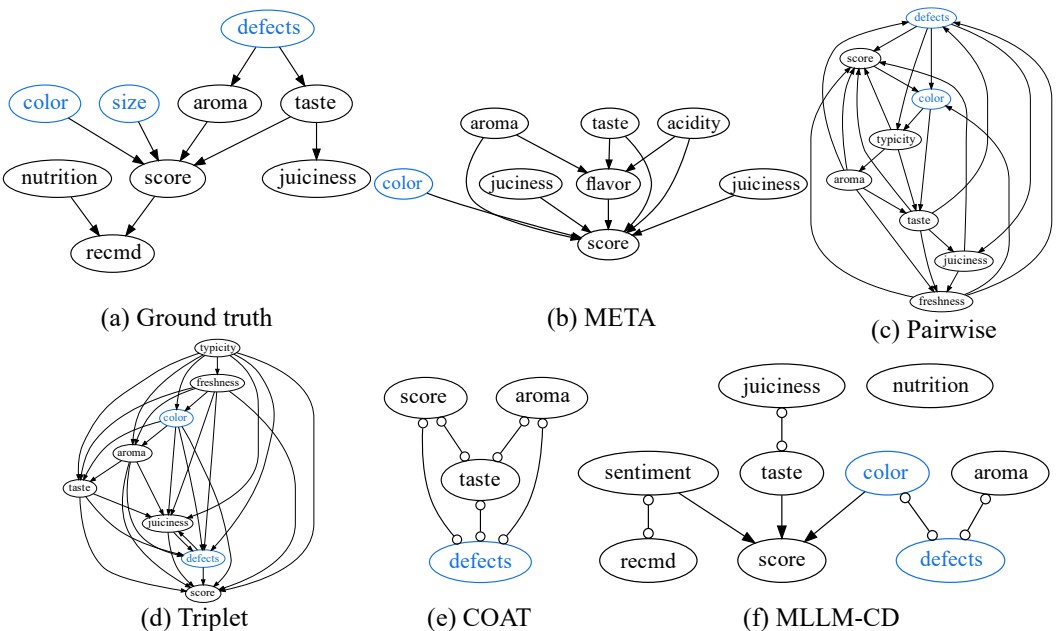

Figure 9: Causal graphs discovered with Gemini 2.0 on the MAG dataset.

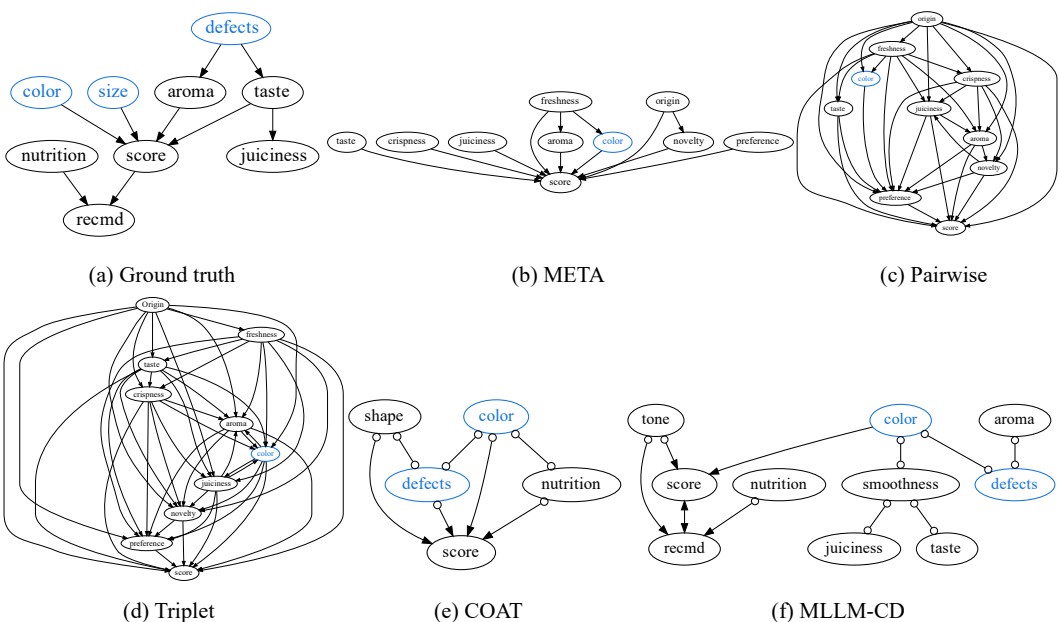

Figure 10: Causal graphs discovered with GPT-4o on the MAG dataset.

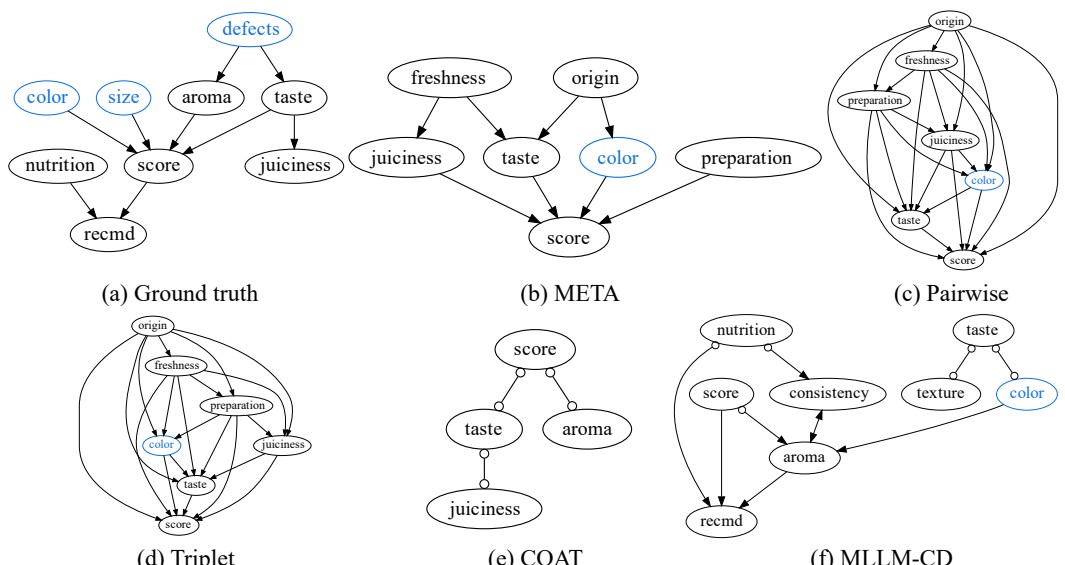

Figure 11: Causal graphs discovered with LLaMA 4 on the MAG dataset.

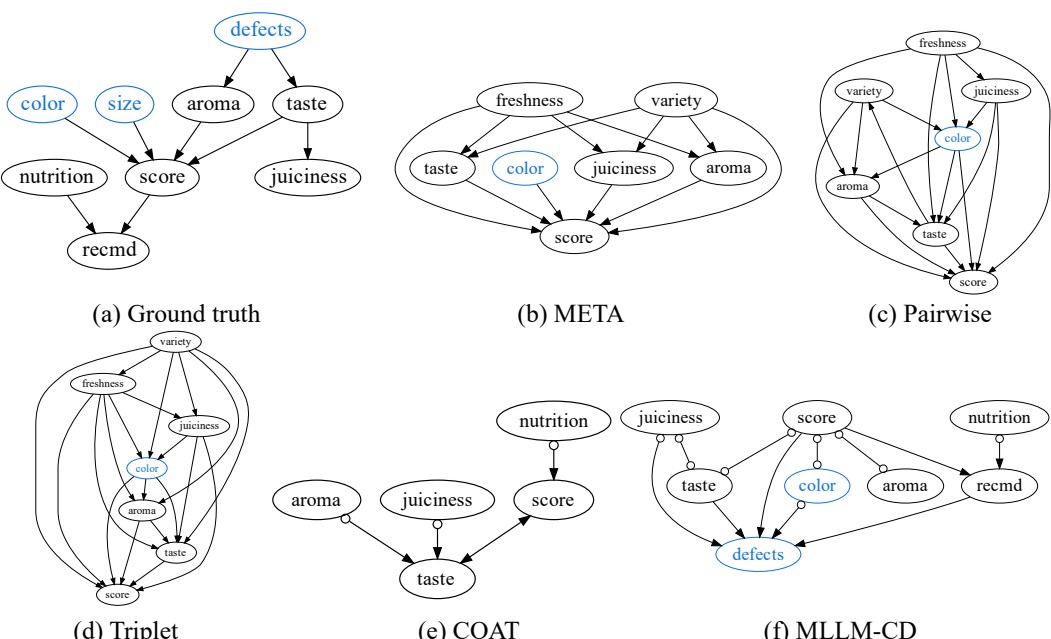

Figure 12: Causal graphs discovered with Grok-2v on the MAG dataset.

### D.5 The Full Results on Lung Cancer Dataset

The detailed results of different methods using various MLLM backbones on the Lung Cancer dataset are presented in Table 7. Consistent with the findings on the MAG dataset, MLLM-CD demonstrates superior performance in both causal factor identification and causal structure discovery in most cases. Qualitative comparisons of discovered causal graphs are shown in Figures 13, 14, 15 and 16. Among all methods, MLLM-CD consistently produces high-quality causal graphs that closely align with the ground truth and the faithful causal graphs illustrated in Figure 7 (b), for example, see Figure 13 (f).

Table 7: Causal factor identification and structure discovery performance on the Lung Cancer dataset.[7]

| LLM | Method | NP ↑ | NR ↑ | NF ↑ | AP ↑ | AR ↑ | AF ↑ | ESHD ↓ |
|---|---|---|---|---|---|---|---|---|
| GPT-4o | META | $0.33 \pm 0.05$ | $0.60 \pm 0.00$ | $0.42 \pm 0.04$ | $\mathbf{1.00 \pm 0.00}$ | $0.50 \pm 0.00$ | $0.67 \pm 0.00$ | $21.67 \pm 4.62$ |
| | Pairwise | - | - | - | $0.67 \pm 0.00$ | $0.50 \pm 0.00$ | $0.57 \pm 0.00$ | $36.33 \pm 10.21$ |
| | Triplet | - | - | - | $0.67 \pm 0.00$ | $0.50 \pm 0.00$ | $0.57 \pm 0.00$ | $47.33 \pm 14.98$ |
| | COAT | $0.47 \pm 0.18$ | $0.40 \pm 0.00$ | $0.42 \pm 0.07$ | $0.33 \pm 0.58$ | $0.08 \pm 0.14$ | $0.13 \pm 0.23$ | $11.67 \pm 3.06$ |
| | MLLM-CD | $\mathbf{0.89 \pm 0.10}$ | $\mathbf{1.00 \pm 0.00}$ | $\mathbf{0.94 \pm 0.05}$ | $0.92 \pm 0.14$ | $\mathbf{0.58 \pm 0.14}$ | $\mathbf{0.69 \pm 0.05}$ | $\mathbf{5.00 \pm 0.00}$ |
| Gemini 2.0 | META | $0.43 \pm 0.07$ | $0.73 \pm 0.12$ | $0.54 \pm 0.08$ | $0.92 \pm 0.14$ | $0.67 \pm 0.14$ | $0.76 \pm 0.10$ | $16.00 \pm 0.00$ |
| | Pairwise | - | - | - | $0.56 \pm 0.10$ | $0.67 \pm 0.14$ | $0.59 \pm 0.02$ | $33.33 \pm 3.51$ |
| | Triplet | - | - | - | $0.56 \pm 0.10$ | $0.67 \pm 0.14$ | $0.59 \pm 0.02$ | $40.00 \pm 8.19$ |
| | COAT | $0.75 \pm 0.25$ | $0.47 \pm 0.12$ | $0.56 \pm 0.11$ | $\mathbf{1.00 \pm 0.00}$ | $0.33 \pm 0.14$ | $0.49 \pm 0.15$ | $8.67 \pm 2.08$ |
| | MLLM-CD | $\mathbf{0.94 \pm 0.10}$ | $\mathbf{1.00 \pm 0.00}$ | $\mathbf{0.97 \pm 0.05}$ | $0.92 \pm 0.14$ | $\mathbf{0.83 \pm 0.14}$ | $\mathbf{0.87 \pm 0.13}$ | $\mathbf{4.67 \pm 0.58}$ |
| LLaMA 4 | META | $0.41 \pm 0.08$ | $0.67 \pm 0.12$ | $0.50 \pm 0.04$ | $0.72 \pm 0.25$ | $0.42 \pm 0.14$ | $0.52 \pm 0.17$ | $19.33 \pm 5.03$ |
| | Pairwise | - | - | - | $0.76 \pm 0.21$ | $0.58 \pm 0.14$ | $0.63 \pm 0.05$ | $32.33 \pm 17.24$ |
| | Triplet | - | - | - | $0.61 \pm 0.10$ | $0.58 \pm 0.14$ | $0.58 \pm 0.02$ | $36.67 \pm 20.40$ |
| | COAT | $0.81 \pm 0.17$ | $0.47 \pm 0.12$ | $0.58 \pm 0.08$ | $\mathbf{1.00 \pm 0.00}$ | $0.25 \pm 0.00$ | $0.40 \pm 0.00$ | $7.67 \pm 0.58$ |
| | MLLM-CD | $\mathbf{0.82 \pm 0.02}$ | $\mathbf{0.93 \pm 0.12}$ | $\mathbf{0.87 \pm 0.06}$ | $0.92 \pm 0.14$ | $\mathbf{0.67 \pm 0.14}$ | $\mathbf{0.76 \pm 0.10}$ | $\mathbf{5.67 \pm 0.58}$ |
| Grok-2v | META | $0.29 \pm 0.04$ | $0.40 \pm 0.00$ | $0.33 \pm 0.03$ | $\mathbf{1.00 \pm 0.00}$ | $0.25 \pm 0.00$ | $0.40 \pm 0.00$ | $20.00 \pm 5.29$ |
| | Pairwise | - | - | - | $\mathbf{1.00 \pm 0.00}$ | $0.25 \pm 0.00$ | $0.40 \pm 0.00$ | $28.00 \pm 8.89$ |
| | Triplet | - | - | - | $\mathbf{1.00 \pm 0.00}$ | $0.25 \pm 0.00$ | $0.40 \pm 0.00$ | $31.00 \pm 8.00$ |
| | COAT | $0.56 \pm 0.21$ | $0.47 \pm 0.23$ | $0.51 \pm 0.22$ | $0.67 \pm 0.58$ | $0.17 \pm 0.14$ | $0.27 \pm 0.23$ | $9.67 \pm 1.53$ |
| | MLLM-CD | $\mathbf{0.87 \pm 0.12}$ | $\mathbf{0.80 \pm 0.00}$ | $\mathbf{0.83 \pm 0.05}$ | $\mathbf{1.00 \pm 0.00}$ | $0.25 \pm 0.00$ | $0.40 \pm 0.00$ | $\mathbf{6.00 \pm 1.00}$ |
| Average | META | $0.36 \pm 0.08$ | $0.60 \pm 0.15$ | $0.45 \pm 0.09$ | $0.91 \pm 0.17$ | $0.46 \pm 0.18$ | $0.59 \pm 0.17$ | $19.25 \pm 4.27$ |
| | Pairwise | - | - | - | $0.74 \pm 0.20$ | $0.50 \pm 0.18$ | $0.55 \pm 0.10$ | $32.50 \pm 9.97$ |
| | Triplet | - | - | - | $0.71 \pm 0.19$ | $0.50 \pm 0.18$ | $0.54 \pm 0.08$ | $38.75 \pm 13.36$ |
| | COAT | $0.65 \pm 0.23$ | $0.45 \pm 0.12$ | $0.52 \pm 0.13$ | $0.75 \pm 0.45$ | $0.21 \pm 0.14$ | $0.32 \pm 0.21$ | $9.42 \pm 2.31$ |
| | MLLM-CD | $\mathbf{0.88 \pm 0.09}$ | $\mathbf{0.93 \pm 0.10}$ | $\mathbf{0.90 \pm 0.07}$ | $\mathbf{0.94 \pm 0.11}$ | $\mathbf{0.58 \pm 0.25}$ | $\mathbf{0.68 \pm 0.19}$ | $\mathbf{5.33 \pm 0.78}$ |

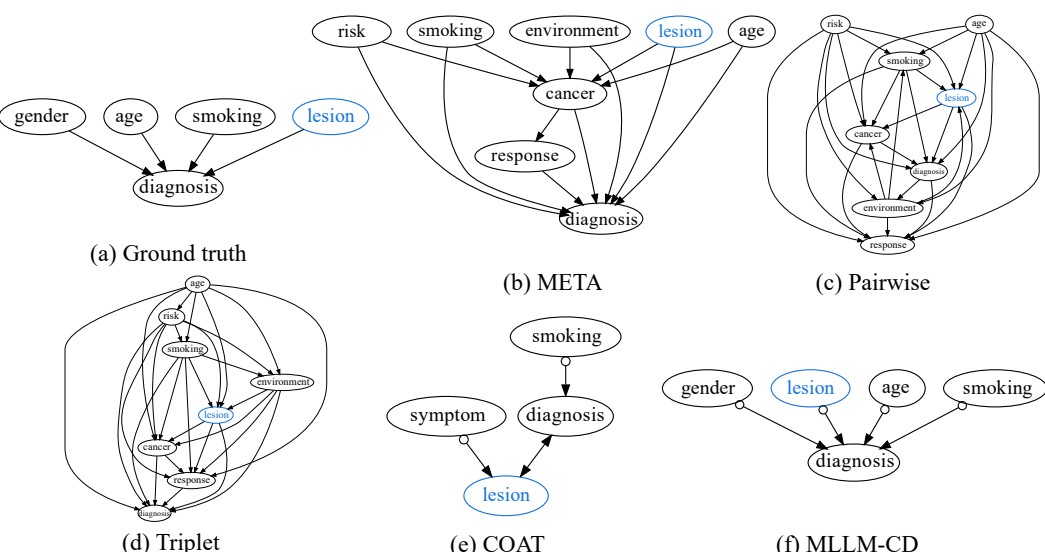

(a) Ground truth     (b) META     (c) Pairwise

(d) Triplet     (e) COAT     (f) MLLM-CD

Figure 13: Causal graphs discovered with Gemini 2.0 on the Lung Cancer dataset.

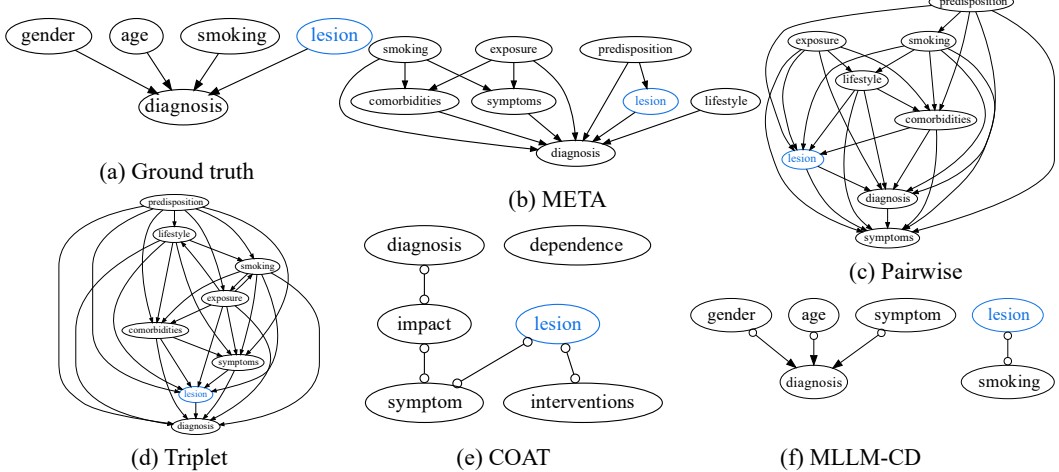

Figure 14: Causal graphs discovered with GPT-4o on the Lung Cancer dataset.

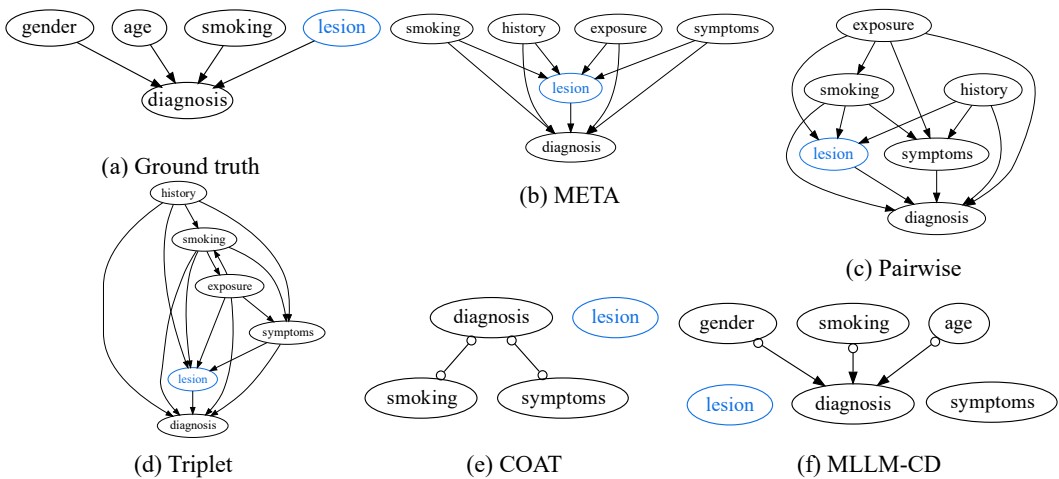

Figure 15: Causal graphs discovered with LLaMA 4 on the Lung Cancer dataset.

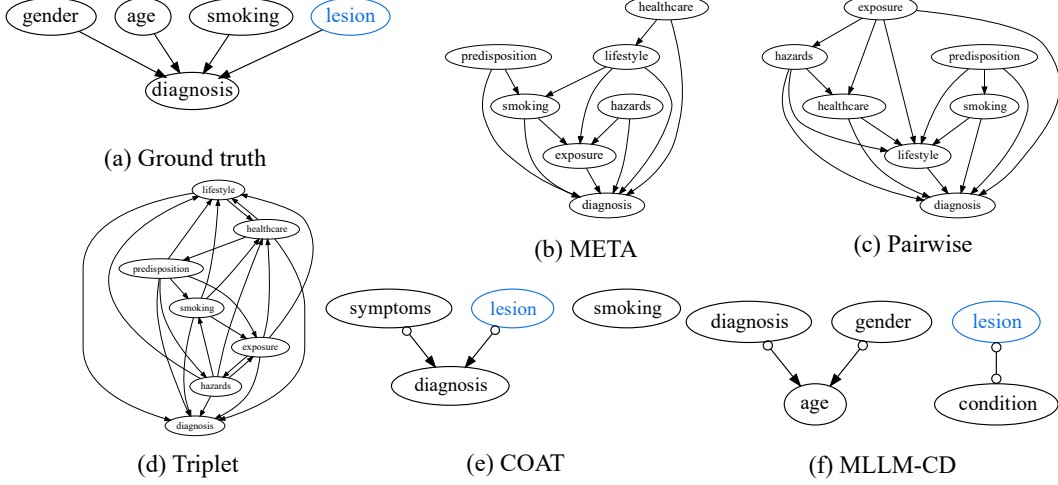

Figure 16: Causal graphs discovered with Grok-2v on the Lung Cancer dataset.

## D.6 The Full Results on Ablation Study

As a complement to Section 4.4, we present additional ablation study results on the MAG and Lung Cancer datasets using two representative MLLMs, i.e., GPT-4o and Gemini 2.0. The detailed results are shown in Table 8 and Table 9, respectively.

Table 8: Ablation study of MLLM-CD on the MAG dataset.

| LLM | Method | NP ↑ | NR ↑ | NF ↑ | AP ↑ | AR ↑ | AF ↑ | ESHD ↓ |
|---|---|---|---|---|---|---|---|---|
| GPT-4o | w/o CFD | $0.83 \pm 0.15$ | $0.67 \pm 0.11$ | $0.73 \pm 0.10$ | $0.63 \pm 0.32$ | $0.22 \pm 0.11$ | $0.30 \pm 0.10$ | $17.00 \pm 2.00$ |
| | w/o CR | $\mathbf{0.87 \pm 0.14}$ | $0.74 \pm 0.17$ | $0.80 \pm 0.15$ | $0.68 \pm 0.30$ | $0.30 \pm 0.23$ | $0.36 \pm 0.20$ | $16.00 \pm 2.65$ |
| | w/o Both | $0.79 \pm 0.26$ | $0.48 \pm 0.32$ | $0.58 \pm 0.34$ | $0.30 \pm 0.26$ | $0.15 \pm 0.13$ | $0.20 \pm 0.17$ | $16.33 \pm 2.08$ |
| | MLLM-CD | $0.83 \pm 0.11$ | $\mathbf{0.85 \pm 0.06}$ | $\mathbf{0.84 \pm 0.09}$ | $\mathbf{0.69 \pm 0.05}$ | $\mathbf{0.41 \pm 0.06}$ | $\mathbf{0.51 \pm 0.04}$ | $\mathbf{15.33 \pm 2.31}$ |
| Gemini 2.0 | w/o CFD | $\mathbf{0.94 \pm 0.10}$ | $0.60 \pm 0.06$ | $0.73 \pm 0.07$ | $0.75 \pm 0.23$ | $0.37 \pm 0.13$ | $0.47 \pm 0.09$ | $15.00 \pm 0.00$ |
| | w/o CR | $0.84 \pm 0.08$ | $0.78 \pm 0.11$ | $0.81 \pm 0.09$ | $0.74 \pm 0.07$ | $0.41 \pm 0.06$ | $0.52 \pm 0.06$ | $15.67 \pm 3.51$ |
| | w/o Both | $0.81 \pm 0.17$ | $0.41 \pm 0.06$ | $0.54 \pm 0.08$ | $\mathbf{1.00 \pm 0.00}$ | $0.26 \pm 0.06$ | $0.41 \pm 0.08$ | $16.33 \pm 2.08$ |
| | MLLM-CD | $0.86 \pm 0.05$ | $\mathbf{0.89 \pm 0.00}$ | $\mathbf{0.87 \pm 0.03}$ | $0.76 \pm 0.08$ | $\mathbf{0.52 \pm 0.13}$ | $\mathbf{0.60 \pm 0.06}$ | $\mathbf{14.00 \pm 3.46}$ |

Table 9: Ablation study of MLLM-CD on the Lung Cancer dataset.

| LLM | Method | NP ↑ | NR ↑ | NF ↑ | AP ↑ | AR ↑ | AF ↑ | ESHD ↓ |
|---|---|---|---|---|---|---|---|---|
| GPT-4o | w/o CFD | $0.61 \pm 0.05$ | $0.73 \pm 0.12$ | $0.66 \pm 0.06$ | $0.33 \pm 0.58$ | $0.17 \pm 0.29$ | $0.22 \pm 0.38$ | $9.67 \pm 1.53$ |
| | w/o CR | $0.83 \pm 0.00$ | $\mathbf{1.00 \pm 0.00}$ | $0.91 \pm 0.00$ | $0.72 \pm 0.25$ | $0.42 \pm 0.14$ | $0.49 \pm 0.09$ | $5.33 \pm 0.58$ |
| | w/o Both | $0.56 \pm 0.10$ | $0.53 \pm 0.31$ | $0.52 \pm 0.22$ | $0.33 \pm 0.58$ | $0.17 \pm 0.29$ | $0.22 \pm 0.38$ | $9.67 \pm 2.08$ |
| | MLLM-CD | $\mathbf{0.89 \pm 0.10}$ | $\mathbf{1.00 \pm 0.00}$ | $\mathbf{0.94 \pm 0.05}$ | $\mathbf{0.92 \pm 0.14}$ | $\mathbf{0.58 \pm 0.14}$ | $\mathbf{0.69 \pm 0.05}$ | $\mathbf{5.00 \pm 0.00}$ |
| Gemini 2.0 | w/o CFD | $0.65 \pm 0.09$ | $0.60 \pm 0.00$ | $0.62 \pm 0.04$ | $\mathbf{1.00 \pm 0.00}$ | $0.33 \pm 0.14$ | $0.36 \pm 0.34$ | $8.00 \pm 1.00$ |
| | w/o CR | $0.89 \pm 0.10$ | $\mathbf{1.00 \pm 0.00}$ | $0.94 \pm 0.05$ | $0.83 \pm 0.29$ | $0.25 \pm 0.00$ | $0.38 \pm 0.04$ | $5.33 \pm 1.53$ |
| | w/o Both | $0.58 \pm 0.14$ | $0.53 \pm 0.12$ | $0.55 \pm 0.11$ | $0.33 \pm 0.58$ | $0.08 \pm 0.14$ | $0.13 \pm 0.23$ | $9.67 \pm 2.31$ |
| | MLLM-CD | $\mathbf{0.94 \pm 0.10}$ | $\mathbf{1.00 \pm 0.00}$ | $\mathbf{0.97 \pm 0.05}$ | $0.92 \pm 0.14$ | $\mathbf{0.83 \pm 0.14}$ | $\mathbf{0.87 \pm 0.13}$ | $\mathbf{4.67 \pm 0.58}$ |

## D.7 MLLM-CD with Different Causal Discovery Algorithms

Since MLLM-CD is designed as a general framework for causal discovery from multimodal unstructured data, it can be implemented with various causal discovery algorithms. In this section, we evaluate the performance of MLLM-CD when combined with different algorithms and compare its effectiveness against the state-of-the-art unstructured causal discovery method, COAT. We select three representative causal discovery algorithms from different categories: (1) PC [31] (constraint-based), (2) GES [105] (score-based), and (3) CAM-UV [106] (functional causal model-based with latent variable handling). Using Gemini 2.0 as the MLLM backbone, we evaluate the structure discovery performance on both the MAG and Lung Cancer datasets. The results, presented in Table 10, show that MLLM-CD achieves overall superior performance than COAT across various causal discovery algorithms, demonstrating its robustness and adaptability.

## D.8 Parameter Analysis

In this section, we analyze the impact of different parameters on the performance of MLLM-CD. We mainly focus on the following parameters: (1) the number of iterations $T$; (2) the number of pairs $K$; (3) the semantic plausibility threshold $\tau_{\text{sem}}$; and (4) the causal consistency threshold $\tau_{\text{causal}}$.

**Iteration Number $T$.** Following [20], we set the maximum number of iterations to 3. As shown in Table 11, the performance of MLLM-CD improves notably from 1 to 2 iterations, while it remains relatively stable between 2 and 3 iterations. In most cases, using 2 iterations provides a good trade-off between performance and computational efficiency.

Figure 17 further illustrates the structural refinement achieved from iteration 1 to iteration 2. Notably, the structure becomes more accurate and less ambiguous, yielding results that more closely align with the faithful causal graph identified by FCI, as shown in Figure 7 (b).

**Pair Number $K$.** We explore the impact of the number of contrastive pairs $K$ on the performance of MLLM-CD. As shown in Figures 18 (a) and 19 (a), the performance of MLLM-CD improves

Table 10: Causal structure discovery results of COAT and MLLM-CD with different causal discovery algorithms on the MAG and Lung Cancer datasets.

| Dataset | CD | Method | AP ↑ | AR ↑ | AF ↑ | ESHD ↓ |
|---|---|---|---|---|---|---|
| MAG | PC | COAT | **1.00** | 0.22 | 0.36 | 19.00 |
| | | MLLM-CD | 0.89 | **0.56** | **0.68** | **14.00** |
| | GES | COAT | 0.67 | 0.22 | 0.33 | 16.00 |
| | | MLLM-CD | **0.75** | **0.33** | **0.46** | **16.00** |
| | CAM-UV | COAT | **1.00** | 0.11 | 0.20 | 16.00 |
| | | MLLM-CD | 0.50 | **0.22** | **0.31** | **13.00** |
| Lung | PC | COAT | **1.00** | 0.50 | 0.67 | 8.00 |
| | | MLLM-CD | 0.75 | **0.75** | **0.75** | **2.00** |
| | GES | COAT | **1.00** | 0.25 | 0.40 | 11.00 |
| | | MLLM-CD | 0.75 | **0.75** | **0.75** | **3.00** |
| | CAM-UV | COAT | **1.00** | 0.25 | 0.40 | 10.00 |
| | | MLLM-CD | 0.60 | **0.75** | **0.67** | **3.00** |

Table 11: The performance of MLLM-CD with different iteration numbers on the MAG and Lung Cancer datasets.

| Dataset | $T$ | NP ↑ | NR ↑ | NF ↑ | AP ↑ | AR ↑ | AF ↑ | ESHD ↓ |
|---|---|---|---|---|---|---|---|---|
| MAG | 1 | 0.84 | 0.78 | 0.81 | 0.74 | 0.41 | 0.52 | 15.67 |
| | 2 | 0.86 | **0.89** | 0.87 | 0.76 | **0.52** | **0.60** | 14.00 |
| | 3 | **0.89** | **0.89** | **0.89** | **0.80** | 0.44 | 0.57 | **13.00** |
| Lung | 1 | 0.89 | 1.00 | 0.94 | 0.83 | 0.25 | 0.38 | 5.33 |
| | 2 | **0.94** | 1.00 | **0.97** | 0.92 | **0.83** | **0.87** | 4.67 |
| | 3 | **0.94** | 1.00 | **0.97** | **1.00** | 0.75 | 0.86 | **4.00** |

with increasing $K$ at first, but it diminishes slightly after reaching a certain point. This implies that using too few pairs may fail to capture sufficient contrastive information for effective factor discovery, while an excessive number of pairs could dilute the guidance signal and reduce model focus. Based on our findings, we choose $K = 5$ as a balanced and effective setting for both datasets and adopt this value in all experiments.

**Semantic Plausibility Threshold** $\tau_{\text{sem}}$. This parameter determines how many counterfactual samples are retained based on their semantic plausibility. A high threshold risks discarding many potentially useful samples, thereby limiting the benefits of counterfactual reasoning in causal discovery. Conversely, a low threshold may admit low-quality or hallucinated samples, introducing noise. As shown in Figures 18 (b) and 19 (b), MLLM-CD achieves optimal and stable performance when $\tau_{\text{sem}} = 0.7$ on both datasets. We therefore set this value in our experiments. In addition, we observe that MLLM-CD is relatively insensitive to variations in $\tau_{\text{sem}}$, indicating robustness to this parameter.

**Causal Consistency Threshold** $\tau_{\text{causal}}$. This parameter controls the stringency of the causal consistency check. As observed in Figures 18 (c) and 19 (c), setting a low threshold results in overly strict filtering, which can reduce the number of usable counterfactuals and hinder performance. On the other hand, a high threshold may allow too many causally inconsistent samples, introducing noise. We find that $\tau_{\text{causal}} = 0.4$ offers a strong performance regarding factor identification and causal structure discovery across both datasets, and we use this value in our experiments.

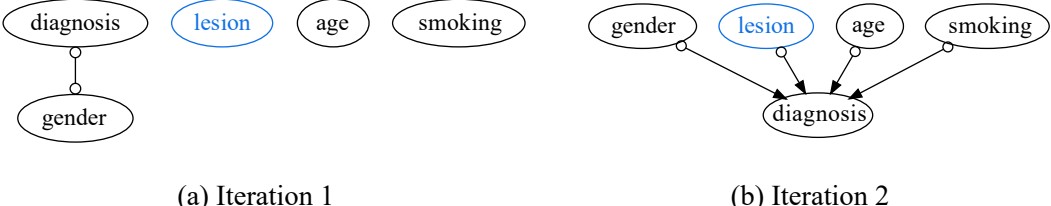

(a) Iteration 1                 (b) Iteration 2

Figure 17: Causal graphs discovered with Grok-2v on the Lung Cancer dataset.

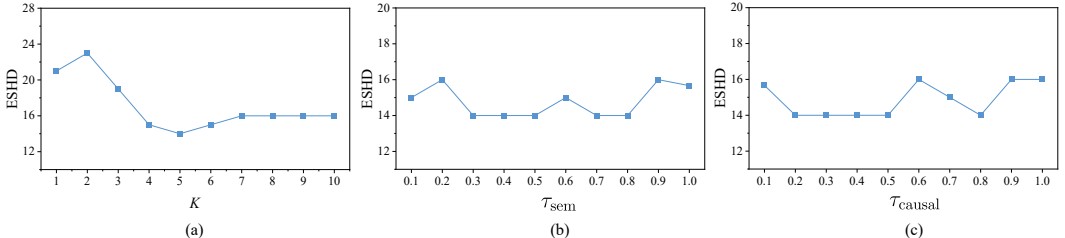

Figure 18: The performance of MLLM-CD with different parameters of $K$, $\tau_{\text{sem}}$, and $\tau_{\text{causal}}$ on the MAG dataset. (The lower the ESHD, the better the performance.)

### D.9 Discussion on the Time Complexity and Scalability

Assume there are $n$ samples, $m$ modalities, $d$ factors, $K$ contrastive pairs, and $T$ iterations. Assume the cost of a single MLLM query as $C_\Psi$ and an image generation query as $C_\Phi$. The number of samples can grow in each iteration due to counterfactual augmentation; let $N_t$ be the number of samples in the $t$-th iteration. We first analyze the complexity of one iteration $t$.

**Contrastive Factor Discovery:** In intra-modal contrastive exploration, it requires $O(m \cdot N_t^2)$ to find top-$K$ intra-modal contrastive pairs, and $O(m \cdot K \cdot C_\Psi)$ for the MLLM query. In inter-modal contrastive exploration, it requires $O(N_t^2)$ for pair selection and $O(K \cdot C_\Psi)$ for the MLLM query. It also requires $O(C_\Psi)$ for factor consolidation and $O(N_t \cdot C_\Psi)$ for factor annotation. Therefore, the total complexity for this module is $O(m \cdot N_t^2 + m \cdot K \cdot C_\Psi + N_t \cdot C_\Psi)$.

**Causal Structure Discovery:** The complexity depends heavily on the chosen causal discovery algorithm $\mathcal{C}$, $N_t$, and $d$. We denote the complexity as $O(C_{\text{CSD}}(N_t, d))$.

**Multimodal Counterfactual Generation:** It requires $O(N_{\text{CF}_t} \cdot (C_\Psi + C_\Phi))$ for counterfactual generation, where $N_{\text{CF}_t}$ is the number of generated counterfactual samples in $t$-th iteration. For semantic plausibility validation and causal consistency validation, it typically requires $O(N_{\text{CF}_t} \cdot m)$ and $O(N_{\text{CF}_t} \cdot d)$. Thus, the total complexity for this module is $O(N_{\text{CF}_t} \cdot (C_\Psi + C_\Phi + d))$.

**Overall Complexity:** Let $N_{max}$ be the maximum $N_t$, we have the following rough upper bound for the time complexity of the overall MLLM-CD:

$$O(T \cdot (mN_{max}^2 + N_{max}C_\Psi + C_{\text{CSD}}(N_{max}, d) + N_{\text{CF}_t} \cdot (C_\Psi + C_\Phi + d))). \tag{D.7}$$

**Scalability:** The scalability of MLLM-CD with respect to the dataset size is primarily challenged by two aspects:

1) The CFD module exhibits a quadratic complexity ($O(mN_{max}^2)$), which can become computationally expensive for large datasets. However, this can be mitigated by using strategies such as negative sampling [107, 108] for contrastive pair selection.

2) The statistical causal discovery algorithm often involves a polynomial complexity in $N_{max}$ and exponential complexity in $d$. Learning causal structures over a large number of samples and factors remains an open problem in the literature [109].

In the future, as we briefly discussed in B.1, we will construct larger-scale multimodal unstructured datasets for causal discovery and improve the scalability and efficiency of MLLM-CD by leveraging advanced sampling techniques [107, 108] and more efficient causal discovery algorithms, such as ENCO [109] and ETCD [110].

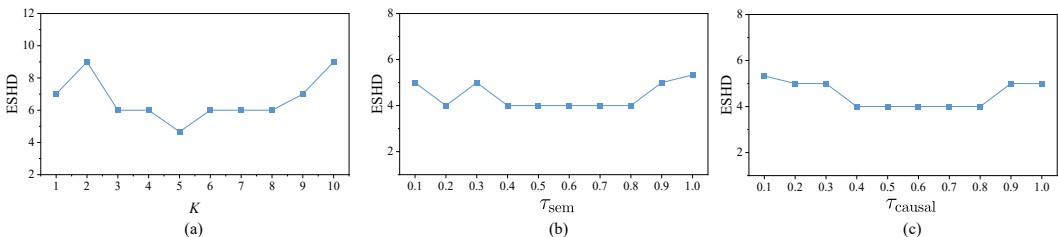

Figure 19: The performance of MLLM-CD with different parameters of $K$, $\tau_{\text{sem}}$, and $\tau_{\text{causal}}$ on the Lung Cancer dataset. (The lower the ESHD, the better the performance.)

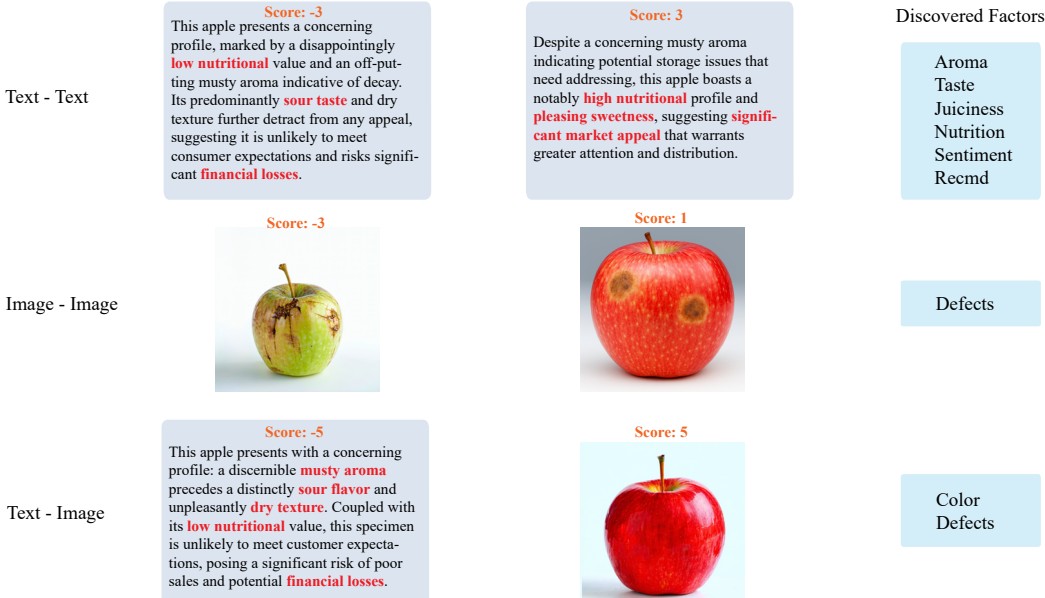

Figure 20: Examples of selected intra-modal and inter-modal contrastive pairs on the MAG dataset.

## D.10 Case Study

We present a case study to illustrate how MLLM-CD contributes to the multimodal unstructured causal discovery process.

**Contrastive Factor Discovery:** We first show some examples of intra-modal and inter-modal contrastive pairs selected by MLLM-CD on the MAG dataset in Figure 20. For intra-modal contrastive exploration, MLLM-CD effectively constructs contrastive pairs that highlight several key variations, such as the differences in nutritional profiles, taste, and the presence of defects. By analyzing the underlying intra-modal interactions, the model successfully identifies a largely accurate and comprehensive set of factors. For inter-modal contrastive exploration, MLLM-CD selects mismatched pairs to expose contradictions between modalities. This encourages the model to uncover more subtle factors governing inter-modal interactions. Notably, it identifies the factor `color`, which was not captured by the intra-modal analysis. This demonstrates the strength and necessity of the CFD module in enhancing factor identification from multimodal unstructured data.

**Multimodal Counterfactual Reasoning:** To demonstrate the effectiveness of the MCR module in generating meaningful counterfactual samples, we present both textual and visual counterfactuals generated by MLLM-CD on the MAG and Lung Cancer datasets, as shown in Figures 21 and 22, respectively. The generated counterfactuals are both semantically plausible and causally consistent. For instance, in Figure 21, when reasoning about the counterfactual scenario "*what if nutritional profile were different*", the model generates a new review comment that aligns semantically and causally with the original context. More importantly, it captures the latent causal relationship between the nutritional profile and market potential (as shown in the ground truth causal graph in Figure 5(a))

Original                                                      Counterfactuals

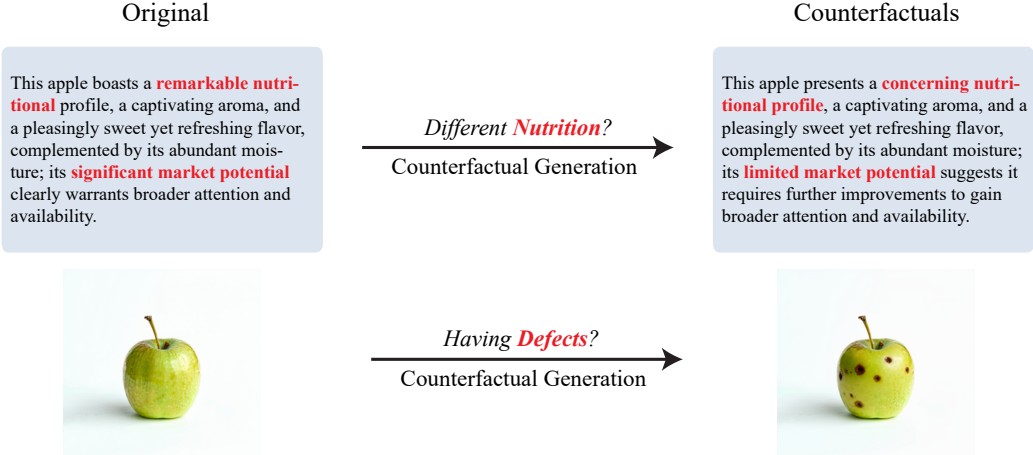

Figure 21: Examples of counterfactual samples generated by MLLM-CD on the MAG dataset.

Original                                                      Counterfactuals

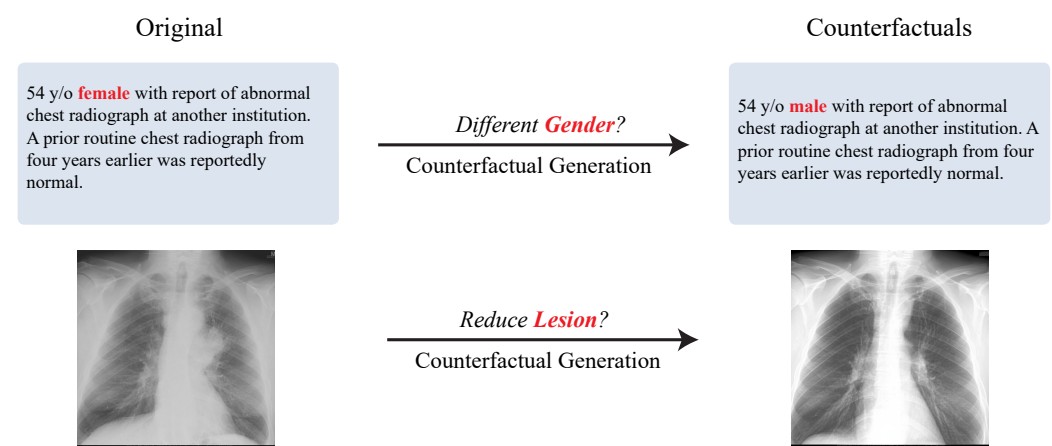

Figure 22: Examples of counterfactual samples generated by MLLM-CD on the Lung Cancer dataset.

based on its knowledge and appropriately modifies the market potential in response to changes in the nutritional profile. Similarly, in the visual domain, the counterfactual samples accurately reflect causal changes. In Figure 22, for instance, the generated image clearly shows a reduction in lesion area, reflecting the intended counterfactual condition. These results highlight the capability of the MCR module to produce high-quality, causally informative counterfactual samples that contribute meaningfully to the causal discovery process.

**Overall Causal Discovery Process:** We further illustrate the overall causal discovery process of MLLM-CD using an example from the MAG dataset, as shown in Figure 23. Given a multimodal, unstructured input, MLLM-CD first identifies the factors `nutrition`, `color`, `defects`, `aroma`, `taste`, `juiciness`, `sentiment`, `recmd`, along with the target variable `score`. For clarity, some of these factors are highlighted directly within the original sample. The model then annotates the values of these factors to construct the structured data as input for causal structure discovery. Once an initial causal graph is inferred, MLLM-CD applies the MCR module for iterative refinement, enhancing the accuracy and reducing structural ambiguity.

# E   Broader Impacts

This work aims to advance causal discovery by extending it to multimodal unstructured data, leveraging the understanding and reasoning capabilities of multimodal large language models (MLLMs). The goal is to enable broader applications and societal benefits, such as accelerating scientific dis-

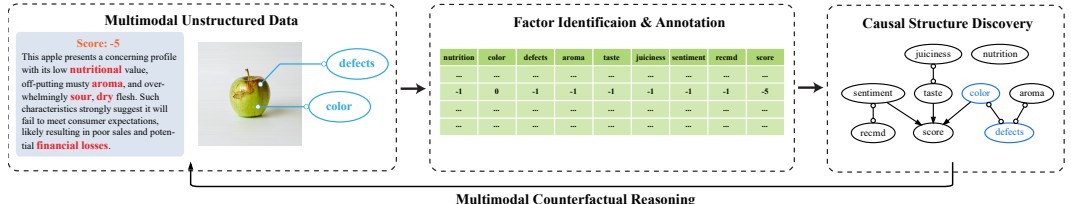

Figure 23: An example of the causal discovery process of MLLM-CD on the MAG dataset.

covery, enhancing multimodal decision-making, and enhancing medical diagnosis systems. This study does not involve any human subjects or raise new ethical concerns. Dataset usage adheres to public availability and anonymization principles. The proposed method is intended for beneficial applications and does not introduce specific risks related to harmful insights, privacy, security, legal compliance, or research integrity, beyond the general considerations for MLLM technologies.

