# OpenReview forum: "Revealing Multimodal Causality with Large Language Models"
_NeurIPS.cc/2025/Conference — NeurIPS 2025 poster_

### Official Review · Reviewer_ynPz · 2025-07-03

**Clarity:** 3
**Significance:** 3
**Originality:** 3
**Rating:** 5
**Confidence:** 4

**Summary:**

The paper introduces **MLLM-CD**, a framework that leverages multimodal large language models (MLLMs) to perform causal discovery directly from unstructured, multimodal data. Key ideas include:

1. **Contrastive Factor Discovery (CFD)** selects intra- and inter-modal contrastive pair samples and prompts an MLLM to propose candidate factors. The MLLM then consolidates and annotates the dataset into a structured dataset based on the proposed factor set.
2. Structured data is fed into a causal discovery algorithm, such as FCI, to produce a causal graph
3. **Multimodal Counterfactual Reasoning (MCR)** generates and validates new counterfactual samples (textual and visual) based on the predicted causal graph, using an MLLM to generate new image and text data. The new data is then added to the dataset for the next round of refinement.

Experiments on a synthetic Multimodal Apple Gastronome (MAG) dataset and a real-world Lung Cancer dataset, evaluated across four MLLMs (GPT-4o, Gemini 2.0, LLaMA-4, Grok-2v), show that MLLM-CD outperforms prior methods in both factor identification and structure recovery.

**Questions:**

Questions:

- Have you experimented with other, simpler strategies for sampling data for factor proposals?
- Could you provide evidence to support Assumption D.2?
- Which CLIP checkpoint is being used?

I will increase my score if more evidence supporting CFD is provided.

**Ethical Concerns:**

["NO or VERY MINOR ethics concerns only"]

**Final Justification:**

The additional experiments have empirically addressed my concerns about the validity of the assumptions. Although I still find the theoretical section of the paper somewhat odd and unnecessary, the proposed method is interesting and demonstrates strong empirical performance. Therefore, I am willing to give this paper an accept rating.

**Limitations:**

Yes

**Paper Formatting Concerns:**

I didn't notice anything.

**Quality:**

2

**Strengths And Weaknesses:**

Strengths

- This work addresses the challenge of causal discovery in unstructured multimodal data, a topic that is both important and timely.
- Experiments demonstrate strong performance in both factor identification and structure recovery.
- This MCR module is well-justified both theoretically and empirically.
- Writing is generally clear and well-structured.

Weakness

- My main concern is the CFD module. Introducing an extra MLLM (e.g., CLIP) adds complexity, and its necessity is not well justified:
    - Ablation study (Sec. 4.1) is limited. CFD is compared only with random sampling. Other simple baselines should also be considered, such as:
        - Random sampling from each category of the target variable;
        - Pairing samples with the largest difference in the target variable;
        - Ablate the Intra-modal and Inter-modal independently.
    - Theoretical justification is misleading. Assumption D.2 claims that $V_{CFD}$ is at least as effective, and potentially superior, in terms of a combined measurement of F1-score. This assumption effectively leads to a circular reasoning fallacy, as it presupposes that the factor sets proposed by CFD is better. This assumption needs to be justify empirically.

---

> ### Author Rebuttal · Authors · 2025-07-31
>
> We thank the reviewer for the valuable feedback and helpful suggestions for improving our paper.
>
> **W1.** My main concern is the CFD module. Introducing an extra MLLM (e.g., CLIP) adds complexity, and its necessity is not well justified:
>
> **W1.1.** Ablation study (Sec. 4.1) is limited. CFD is compared only with random sampling. Other simple baselines should also be considered, such as:
>
> * Random sampling from each category of the target variable;
>
> * Pairing samples with the largest difference in the target variable;
>
> * Ablate the Intra-modal and Inter-modal independently.
>
> **RW1.1.** Thank you for your constructive suggestions. We have conducted additional ablation studies to validate the effectiveness of different sampling strategies in terms of their impact on factor discovery (NP, NR, and NF) and overall discovery accuracy (ESHD). The key quantitative results are shown in Tables 1.1 and 1.2, and we have the following observations:
>
> * Compared with random sampling, pairing with the largest difference (simple pair) provides improved ESHD.
> * Incorporating finer-grained signals by finding semantic-level contrastive pairs within each modality, intra-modal further improves the completeness of factor discovery. However, it often overlooks the cross-modal interactions, leaving factors like “defects” or “size” undiscovered.
> * Inter-modal acts as an ideal complement to intra-modal, as it examines across modalities and can effectively identify factors missed by intra-modal. Using inter-modal only has lower quantitative results, as it is prompted to emphasize factors tied to cross-modal interactions, as a complement to intra-modal, rather than aiming to discover all factors.
> * By combining both strategies, our MLLM-CD achieves the best performance in terms of NR, NF, and ESHD.
>
> The full quantitative results and some qualitative comparisons will be provided in the revised manuscript.
>
> **Table 1.1** Ablation study of different sampling strategies on the MAG dataset.
>
> |                | NP↑        | NR↑       | NF↑       | ESHD↓      |
> | -------------- | --------- | -------- | -------- | --------- |
> | Random         | **0.94** | 0.60     | 0.73     | 15.00     |
> | Simple Pair    | 0.79      | 0.70     | 0.75     | 13.67     |
> | Intra-modal | 0.88      | 0.78     | 0.82     | 13.67     |
> | Inter-modal | 0.67      | 0.59     | 0.63     | 15.33     |
> | MLLM-CD        | 0.89      | **0.89** | **0.89** | **13.33** |
>
>
> **Table 1.2** Ablation study of different sampling strategies on the Lung dataset.
>
> |                | NP↑       | NR↑       | NF↑      | ESHD↓     |
> | -------------- | -------- | -------- | -------- | -------- |
> | Random         | 0.65     | 0.60     | 0.62     | 8.00     |
> | Simple Pair    | 0.50     | 0.60     | 0.55     | 6.33     |
> | Intra-modal | 0.80     | 0.80     | 0.80     | 6.00     |
> | Inter-modal | 0.53     | 0.53     | 0.53     | 7.00     |
> | MLLM-CD        | **0.94** | **1.00** | **0.97** | **4.67** |
>
>
> **W1.2.** Theoretical justification is misleading. Assumption D.2 claims that V_{CFD} is at least as effective, and potentially superior, in terms of a combined measurement of F1-score. This assumption effectively leads to a circular reasoning fallacy, as it presupposes that the factor sets proposed by CFD is better. This assumption needs to be justify empirically.
>
> **RW1.2.** Thank you. Apart from existing empirical evidence in the ablation study (Table 3, Page 9), e.g., NF significantly decreases by 36% on the Lung dataset without CFD, we have provided additional empirical comparison between CFD and direct prompting to support this assumption. Key results are shown in Table 1.3. As we can see, CFD consistently outperforms direct prompting in terms of both correctness and completeness on two datasets, and the significance test results (p<0.05) indicate that this large improvement is statistically meaningful. This empirical evidence will be included in the final version of our paper.
>
> **Table 1.3** Comparison between CFD and direct prompting on the factor discovery performance.
>
> |      |         | NP↑        | NR↑         | NF↑         |
> | ---- | ------- | ---------- | ---------- | ---------- |
> | MAG  | Direct  | 0.6945     | 0.4074     | 0.5094     |
> |      | CFD     | **0.8796** | **0.8148** | **0.8453** |
> |      | p-value | 0.0129     | 0.0041     | 0.0023     |
> | Lung | Direct  | 0.6667     | 0.5333     | 0.5926     |
> |      | CFD     | **0.8889** | **1.0000** | **0.9394** |
> |      | p-value | 0.0471     | 0.0099     | 0.0165     |
>
>
>
> **Q1.** Have you experimented with other, simpler strategies for sampling data for factor proposals?
>
> **RQ1.** Thank you. We have now provided the comparison of different sampling strategies for factor proposal in **RW1.1**. In summary, the intra-modal strategy offers finer-grained contrastive guidance and leads to superior discovery performance compared to "random" and "simple pair" methods. The inter-modal strategy, as a complement, can effectively identify factors missed by the Intra-modal method. Combining both strategies achieves the best overall performance.
>
>
> **Q2.** Could you provide evidence to support Assumption D.2?
>
> **RQ2.** Thank you. We have now directly compared the performance of CFD and direct prompting for factor proposal in **RW1.2**. As shown, CFD consistently outperforms direct prompting with statistically meaningful improvements, providing additional empirical evidence in support of Assumption D.2.
>
>
> **Q3.** Which CLIP checkpoint is being used?
>
> **RQ3.** Many thanks. We used the ViT-B/32 checkpoint from OpenAI’s official CLIP release. We will add this detail to Section E.3.1 (Implementation Details, Page 16) in the final version of the paper.

---

> > ### Comment · Reviewer_ynPz · 2025-08-01
> >
> > Thank you for the additional experiments. These results addressed my concerns. I will raise my score accordingly.

---

> > > ### Author Response · Authors · 2025-08-02
> > > **Thank you**
> > >
> > > Dear Reviewer ynPz,
> > >
> > > We are glad to hear that our rebuttal addressed your concerns. These additional results will be included in the final version of our paper. Thank you for raising the score.

---

### Official Review · Reviewer_EBFe · 2025-07-03

**Clarity:** 4
**Significance:** 4
**Originality:** 3
**Rating:** 5
**Confidence:** 3

**Summary:**

This paper proposes MLLM-CD, a framework for causal discovery from multimodal data using MLLMs. The approach consists of three key components: (1) Contrastive Factor Discovery (CFD), which identifies candidate causal variables by comparing semantically similar multimodal instances with different outcomes; (2) Causal Structure Discovery, which uses these factors to infer the causal graph; and (3) Multimodal Counterfactual Reasoning (MCR), which generates and evaluates counterfactual examples to refine the graph structure. The method is evaluated on a synthetic multimodal dataset (MAG) and a real-world lung cancer dataset, using standard metrics for node and edge accuracy, as well as structural Hamming distance. Results are compared against existing baselines in multimodal and causal discovery.

**Questions:**

Given my limited background in causal inference, I would welcome clarification on the following points:

1. If the model generates incorrect or implausible counterfactuals, how might that affect the final causal graph? Could errors at the counterfactual stage mislead the entire structure discovery process?

2. Even with the use of MLLMs, the model can only infer causes based on what is explicitly represented in the text and images. How does the method handle potential unobserved confounders or latent factors that are not captured in the input modalities?

**Ethical Concerns:**

["NO or VERY MINOR ethics concerns only"]

**Final Justification:**

The integration of MLLMs into causal discovery is a timely and promising direction. The concerns I raised regarding the limited evaluation on small, binary-variable datasets and the absence of benchmarks were acknowledged in detail. While these limitations remain unresolved, I appreciate the authors’ justification and their commitment to future contributions in this area. Similarly, while the handling of unobserved confounders remains a broader challenge, the use of the FCI algorithm and contrastive discovery mechanisms provides a reasonable safeguard within scope. The additional explanation of the verification process for counterfactuals was helpful, though I believe further refinement in this area could strengthen the robustness of the method. Given my limited background in causal inference, I maintain a confidence score of 3, in case there are methodological limitations I may have overlooked.

**Limitations:**

yes

**Quality:**

3

**Strengths And Weaknesses:**

## Strengths

1. The overall formulation of the framework is well-structured and logically presented, with a clear separation of its core components. The integration of MLLMs into causal discovery is thoughtfully executed, where its use during counterfactual reasoning for refining causal graphs seems novel and well-motivated.

2. The paper compares the proposed method against a range of baselines, including both traditional and recent MLLM-based approaches.

## Weaknesses

1. The paper evaluates its approach on only two datasets, especially with a relatively small number of causal factors for the Lung Cancer dataset. This could raise questions about the generalizability of the results, especially in more complex or large-scale real-world scenarios.

2. All variables used in the experiments are binary, which simplifies causal discovery but limits the applicability of the method. It remains unclear whether the approach can handle continuous or ordinal variables, or how well it would perform in settings with more diverse data types.

---

> ### Author Rebuttal · Authors · 2025-07-31
>
> We are grateful for the reviewer’s positive feedback and for the thoughtful questions, which help clarify important aspects of our method’s robustness and limitations.
>
> **W1.** The paper evaluates its approach on only two datasets, especially with a relatively small number of causal factors for the Lung Cancer dataset. This could raise questions about the generalizability of the results, especially in more complex or large-scale real-world scenarios.
>
> **RW1.** Thank you. Discovering causality from multimodal unstructured data has significant applications in areas like scientific discovery and healthcare [1]. However, there is currently no public benchmark that can be used for large-scale evaluation.
>
> Our work builds on the pioneering research [2]. Following their experimental practices, we have taken the next step by carefully constructing two new multimodal datasets, one synthetic and one real-world, to validate our proposed method. While these datasets are not large-scale, they are, to our knowledge, the first of their kind and represent a crucial step in benchmarking this new capability.
>
> To be prepared for future large-scale scenarios, we have analyzed the **complexity and scalability** of our method in Section E.9 (Page 24). To handle large-scale datasets, the bottleneck is in contrastive sampling and causal discovery algorithms. As we discussed in Lines 435-437, possible solutions include leveraging efficient sampling techniques [3, 4] and causal discovery algorithms, such as ENCO [5] and ETCD [6].
>
> We acknowledged the limitation of lacking large-scale benchmarks (Appendix B.1, Page 4). We are committed to pushing this important research direction forward and plan to contribute to larger, more diverse public benchmark dataset in future work. We believe this will be a crucial step for the community, and we thank the reviewer for highlighting its importance.
>
>
> **W2.** All variables used in the experiments are binary, which simplifies causal discovery but limits the applicability of the method. It remains unclear whether the approach can handle continuous or ordinal variables, or how well it would perform in settings with more diverse data types.
>
> **RW2.** Thank you for raising this concern. We used binary variables as a common and simplifying starting point [2, 7], which is practical for many real-world factors (e.g., presence/absence of a lesion, smoker/non-smoker).
>
> However, our framework is **not inherently limited to binary variables**. The factor annotation prompt (Appendix E.6, page 15) can be easily adapted to request continuous or categorical values. For example, the prompt can be changed to:
>
> >“*For each sample, assign a continuous score within [0, 1] to each factor based on the criteria above*”.
>
> Then, the causal discovery algorithm can be adjusted to accommodate the new data format. For instance, a constraint-based method would need a conditional independence test suitable for continuous or categorical data. We will add this clarification to Sections 3.2 and 3.3 to highlight the potential for extension.
>
>
> **Q1.** If the model generates incorrect or implausible counterfactuals, how might that affect the final causal graph? Could errors at the counterfactual stage mislead the entire structure discovery process?
>
> **RQ1.** Thank you. We have a crucial safeguard in place to prevent this. As detailed in Section 3.4 (Lines 212-219), every counterfactual sample generated by the MLLM undergoes two validation checks before it is added to our dataset:
>
> 1) **Semantic Plausibility (Eq. 4):** We ensure the generated sample is semantically coherent and not a wild hallucination.
> 2) **Causal Consistency (Eq. 6):** We check if the changes in the counterfactual are consistent with the current causal graph G(t). For instance, an intervention on a node should not affect its non-descendants.
>
> Only counterfactuals that pass both checks are used for refinement. This filtering mechanism is designed specifically to prevent errors from misleading the structure discovery process. However, the threshold parameters need to be tuned empirically, as overly loose validation can introduce noise and harm discovery performance (Appendix E.8, Figures 18&19, Pages 24&25).
>
>
> **Q2.** Even with the use of MLLMs, the model can only infer causes based on what is explicitly represented in the text and images. How does the method handle potential unobserved confounders or latent factors that are not captured in the input modalities?
>
> **RQ2.** This is a fundamental and common challenge in causal discovery [8, 9], though it is currently beyond the main focus of this work. However, our framework alleviates this problem in two ways:
>
> 1) **Factor Discovery:** Our Contrastive Factor Discovery (CFD) module is designed to uncover a more comprehensive set of causal factors from the data, reducing the chances of omitting important variables that could act as confounders.
> 2) **Structure Discovery:** We use the FCI algorithm, which is designed to maintain robust performance in cases with unobserved confounders (Lines 183-184, Page 5). FCI outputs a graph that can explicitly represent uncertainty about latent confounding through bidirected edges. This is a standard and robust approach in the causal discovery literature for dealing with this problem.
>
> In the future, we will explore more challenging and complex scenarios to develop mechanisms with MLLMs that explicitly identify potential confounders to help address this problem.
>
>
> **References:**
>
> [1] R Tu, et al. Neuropathic pain diagnosis simulator for causal discovery algorithm evaluation. NeurIPS, 2019.
>
> [2] C Liu, et al. Discovery of the hidden world with large language models. NeurIPS, 2024.
>
> [3] L Xu, J Lian, et al. Negative sampling for contrastive representation learning: A review. CoRR, 2022.
>
> [4] T Chen, S Kornblith, et al. A simple framework for contrastive learning of visual representations. ICML, 2020.
>
> [5] P Lippe, T Cohen, E Gavves. Efficient neural causal discovery without acyclicity constraints. ICLR, 2022.
>
> [6] X Li, T Liu. Efficient and trustworthy causal discovery with latent variables and complex relations. ICLR, 2025.
>
> [7] A Chen, Q Zhou. Causal Discovery on Dependent Binary Data. AISTATS, 2025.
>
> [8] P Spirtes, C N Glymour, R Scheines. Causation, prediction, and search. MIT Press, 2000.
>
> [9] J M Ogarrio, P Spirtes, J Ramsey. A hybrid causal search algorithm for latent variable models. In the Conference on Probabilistic Graphical Models. 2016: 368-379.

---

> > ### Author Response · Authors · 2025-08-06
> > **Rebuttal follow-up**
> >
> > Dear Reviewer EBFe,
> >
> > Thank you for your efforts in reviewing our work and for your positive feedback. We would like to kindly ask whether our responses have fully addressed your concerns. We appreciate any further feedback and would be happy to answer any additional questions.
> >
> > Thank you once again for your time and consideration.
> >
> > Best regards,
> >
> > The Authors

---

> > ### Comment · Reviewer_EBFe · 2025-08-06
> >
> > Thank you for the detailed response. I believe the verification module could be further improved, and the authors have acknowledged the dataset limitations. However, given the overall contributions and level of excitement, I will maintain my current score.

---

> > > ### Author Response · Authors · 2025-08-06
> > >
> > > Dear Reviewer EBFe,
> > >
> > > Thank you for your follow-up and acknowledgement. We appreciate your thoughtful feedback and will consider your suggestions for future improvements.
> > >
> > > Best regards,
> > >
> > > The Authors

---

### Official Review · Reviewer_yjmf · 2025-07-04

**Clarity:** 3
**Significance:** 3
**Originality:** 2
**Rating:** 3
**Confidence:** 4

**Summary:**

The authors tackle the problem of inferring the underlying causal structure from multimodal unstructured data. They do this by first using multimodal LLMs (MLLM) to extract the causal factors within and across different modalities. This is done via extracting representations for a sample within a modality and computing the cosine similarity within and across modalities. The samples with high misalignment (high cosine distance) are fed along with prompts as contrastive pairs to extract the factors causing these differences, which are then consolidated. A causal structure is then inferred from these factors via a causal discovery algorithm. To improve the causal discovery, counterfactual data is generated from the MLLM for uncertain causal relationships using the internal knowledge of the MLLM. These are checked for semantic plausibility and causal consistency and then appended to the original dataset. The method is tested on two synthetic datasets.

**Questions:**

*Questions regarding counterfactual reasoning module:*
- Does it actually make sense to augment the "counterfactual" sample back into the dataset? For example if the true causal relationship is $X \to Y$, and the causal discovery algorithm returns $X - Y$ then an intervention on $Y$ (which is what you are doing by setting $V_a=v'_{ka}$ (L207)) would make $X$ and $Y$ independent. Putting this back into the sample doesn't actually help learn the relationship $X \to Y$. In fact, it will make the causal discovery (which assumes the whole dataset has come from the same distribution) worse.
- The semantic plausibility does not make sense to me. By definition, a counterfactual sample is from a different distribution and hence the embedding *should* be different from the observational sample. It seems non-trivial to me to quantify how different it should be. For $X \to Y$, intervening on $X$ might not create too different of a sample, but for $X \to Z_1 \to Z_2 \to \cdots \to Z_D \to Y$ intervening on $X$ may create a drastically different sample. Furthermore, is the inequality in equation 4 meant to be less than $\tau$?
- For the causal consistency why isn't equation 6 an equality with $0$? A change should not affect the non-descendents at all. How does this algorithm handle inconsistencies with the learnt graph and the "counterfactual sample" generated by the LLM?
- What if the MLLM does not have information about the query asked as it does not have enough information about a concept? Won't it still generate data however useless it may be?

**Ethical Concerns:**

["NO or VERY MINOR ethics concerns only"]

**Final Justification:**

I still have reservations about this work. I believe the factor discovery with MLLMs is sound and interesting. I have some doubts about the theory and validity of the counterfactual samples: how they are generated (what exact distribution they are generated from), how they are combined with the observational data to feed into the CD algorithm, and what exactly the LLM does when it does not have domain knowledge.

If the authors are not performing interventions to compute “counterfactuals”, it will simply reproduce samples from the observational conditionals. Which should not help identify causal relations.

**Limitations:**

Yes

**Quality:**

2

**Strengths And Weaknesses:**

**Strengths:**
- Although the idea of extracting causal variables from unstructured data using LLMs is not new, using multimodal inputs is definitely an interesting idea.

**Weaknesses:**
- The counterfactual reasoning module is not rigorous enough and I have a few questions regarding the validity of this approach (see questions). It's not clear to me that this module actually provides the benefits that it claims to provide. A more rigorous testing of this can be done in simpler causal discovery settings where the variables are clearly defined. If it is tested rigorously in these settings, it would be reasonable to use it in the harder unstructured data setting. I'm not sure the semantic plausibility and causal consistency are rigorous enough checks on the validity of counterfactual samples.
- The theory makes assumptions that make the conclusion of the theory obvious. In my opinion, the aim of the paper should be to show that these assumptions are true. This is lacking in the current work.
    - Assumption D.2 states that the contrastive probing increases the F1 score of finding the true factors, which is then used to prove that contrastive probing improves structural accuracy.
    - Similarly, assumption D.6 states that the MLLM can generate valid counterfactuals. If this were true, the MLLM would have internalised the correct structure and functional mechanism to determine the whole causal model. Why not just prompt the MLLM to return the correct model, or generate the correct counterfactual data and simply perform regression to find the causal structure.
- Engagement with causal representation is missing, which is the task this paper is trying to solve. This area has conditions (e.g. interventions) under which the causal variables and structure is identifiable [1]. An obvious missing baseline is comparison against these methods in identifiable cases. This would show whether the method works when we know the conditions are there to identify the variables and structure.
- The experiments are only carried out on two different toy datasets, for a method without formal guarantees more experiments with varying conditions would be useful. The different components were not tested thoroughly except for one ablation study. This raises questions like: Does the factor discovery reliably recover the correct factors? Does the counterfactual reasoning actually generate reasonable samples?

[1] https://openreview.net/forum?id=lk2Qk5xjeu

---

> ### Author Rebuttal · Authors · 2025-07-31
>
> Thank you for the detailed and insightful feedback. We hope our response can sufficiently address your concerns.
>
> **W1** Questions regarding MCR and additional test settings.
>
> **RW1** Thank you. We have provided point-by-point responses in **RQ1-RQ3** to address your concerns about the MCR module and validation checks.
>
> Furthermore, we highlight that the validity of MCR has been extensively demonstrated through our experiments:
>
> 1) We provide **direct, quantitative evidence** of MCR's benefits in ablation study (Table 3, Page 9). Removing MCR leads to a clear performance drop, e.g., AF drops from 0.87 to 0.38 on the Lung Cancer dataset.
> 2) Case studies in Appendix E.10 (Figs 21-22, Page 26) provide **qualitative evidence** of causally meaningful and semantically plausible samples generated by MCR, that are crucial for refining the causal graph (Fig 17, Page 24).
>
> We have tested MCR on **standard causal datasets** (Aisa, Child [2]) in Table 1.1. MCR consistently improves structural discovery in classic settings, supporting the rationale for applying it in our unstructured cases.
>
> Table 1.1 Causal discovery results.
> |||AP↑|AR↑|AF↑|SHD↓|
> |-|-|-|-|-|-|
> |Asia|FCI|0.77|0.50|0.58|9.67|
> ||FCI+MCR|**0.89**|**0.58**|**0.67**|**8.33**|
> |Child|FCI| 0.91|0.68|0.78|24.0|
> ||FCI+MCR|**0.93**|**0.74**|**0.82**|**22.0**|
>
> **W2.1** Justification of Assumption D.2
>
> **RW2.1** We justify this from two aspects:
>
> 1) **Mechanistic Design:** CFD approach offers critical guidance beyond direct prompting by encouraging the MLLM to focus on salient differences in inter- and intra-modal sample pairs. This helps uncover genuine causal factors, rather than superficial features. In contrast, direct prompting usually fails to capture comprehensive factors, as in Fig 2.
> 2) **Empirical Evidence:** Our experiments strongly support this assumption: **Tables 1&2** (Page 7, 9) show that MLLM-CD (using CFD) consistently achieves higher NR and NF than COAT (with direct prompting). Similarly, in the ablation study (Table 3, Page 9), removing CFD leads to a 36% drop in NF on the Lung dataset.
>
> To further support this, we include a direct comparison between CFD and direct prompting in Table 2.1. CFD consistently outperforms direct prompting in both correctness and completeness, and a significance test (p<0.05) confirms the improvement is statistically meaningful.
>
> Table 2.1 Comparison between CFD and direct prompting.
> |||NP↑|NR↑|NF↑|
> |-|-|-|-|-|
> |MAG|Direct|0.6945|0.4074|0.5094|
> ||CFD|**0.8796**|**0.8148**|**0.8453**|
> ||p-value|0.0129|0.0041|0.0023|
> |Lung|Direct|0.6667|0.5333|0.5926|
> ||CFD|**0.8889**|**1.0000**|**0.9394**|
> ||p-value|0.0471|0.0099|0.0165|
>
> **W2.2** Justification of Assumption D.6.
>
> **RW2.2** We justify Assumption D.6 as follows:
>
> 1) Assumption D.6 does **not** claim that MLLM provides a perfect causal model. Instead, it makes a weaker and more realistic claim: there exists a **non-zero probability** $p_{MCR}>0$ (Line 146, Page 7) of generating a validated counterfactual. Our method uses these insights while being robust to potential failures through validation checks.
> 2) **Qualitative evidence** supports this: as shown in Figs 21&22 (Page 26), MLLM-generated counterfactuals often reflect correct causal dependencies. For example, lowering “nutrition” also affects “recmd”. Fig 17 (Page 24) further shows a significant improvement by MCR, confirming that $p_{MCR}>0$ is **a practical assumption**.
> 3) **Quantitative support** comes from ablation studies (Tables 7&8). It directly measures the significant benefit of the MCR module (e.g., AF drops from 0.87 to 0.38 when MCR is removed on Lung dataset).
> 4) While MLLMs can aid simple causal tasks [1], they still require guidance with observational data and validation checks to reduce hallucination and inconsistency, as evidenced by baseline results like META in Tables 1&2 (Pages 7&9). Thus, we cannot directly prompt it for the graph or data.
>
> **W3** Engagement with causal representation
>
> **RW3** The task setting in this work is quite different from those used in standard causal representation learning (CRL):
>
> 1) Standard CRL methods usually rely on data from multiple environments (e.g., observational and interventional) as input. Our work addresses the more common case where only unstructured and observational data is available.
> 2) CRL seeks identifiability by recovering a latent vector z mathematically equivalent to true causal variables, which is typically in **latent space and lacks labels** [3]. In contrast, MLLM-CD directly discovers human-interpretable causal factors (e.g., “color”, “lesion”) and then learning the causal graph over these extracted factors.
>
> Thus, it is not directly comparable with CRL in our tasks. We will introduce representative CRL methods and discuss their connections in Section 2.
>
> **W4** Experiment scales and questions on modules
>
> **RW4** Discovering causality from multimodal unstructured data has significant applications in areas like scientific discovery and healthcare [4]. However, no public benchmark datasets exist for large-scale evaluation.
>
> Our work builds on the pioneering study [5]. Following their practices, we carefully constructed two new multimodal datasets, one synthetic (Multimodal Apple Gastronome) and one real-world (Lung Cancer), to validate our method. While these datasets are not large-scale, they are the first of their kind and represent a key step in benchmarking this new capability.
>
> Within the context of these available benchmarks, we extensively evaluated our framework:
>
> 1) **Factor Discovery:** Tables 1&2 (Pages 7, 9) show that MLLM-CD consistently outperforms baselines in NP, NR, and NF, confirming reliable recovery of causal factors.
>
> 2) **Rationale of Counterfactuals:** The benefit of counterfactual reasoning is proven by the significant performance gain in ablation study (Tables 7&8, Page 22). Furthermore, case studies in Appendix E.10 (Figs 21-22, Page 26) offer qualitative evidence of how reasonable samples are generated.
>
> 3) **Thorough Testing:** We provide a comprehensive analysis of the components, including:
>
>    * **Four MLLM backbones** (Tables 5&6, Pages 17, 20).
>    * **Different causal discovery algorithms** (Table 9, Page 23).
>    * **Parameter analysis** in Appendix E.8 (Page 22).
>    * Additional ablation studies on sampling strategies in **RW1.1** to reviewer ynPz.
>
> This comprehensive evaluation highlights the robustness and effectiveness of our method.
>
> We acknowledge the limitation of datasets (Appendix B.1) and are committed to contributing larger benchmarks in future work. We believe this will be a crucial step for the community, and we thank the reviewer for highlighting this direction.
>
> **Q1** Rationale of MCR
>
> **RQ1** Thanks. There may be some misunderstanding, as **MCR fundamentally differs from a statistical hard intervention** (e.g., do-calculus). We do not simply intervene on Y to make X and Y independent. Instead, MCR uses the MLLM’s reasoning and world knowledge to generate a plausible, causally coherent new sample.
>
> Given $G^{(t)}$ and a counterfactual scenario where $V_a$ changes, the MLLM infers how other factors would adjust (Lines 197-198). For example, if we change Y, it evaluates whether X should also change, preserving causal coherence and reinforcing the true direction X->Y.
>
> Thus, rather than harming discovery, we are **implicitly encoding the MLLM’s world knowledge into data**, creating samples that complement observational data and help resolve structural ambiguities.
>
> **Q2** Rationale of semantic plausibility check
>
> **RQ2** We clarify that:
> 1) The motivation of this check is to **against hallucination**. While counterfactuals could differ from the original, their **semantic** embeddings are expected to remain similar as: 1. MLLM is prompted to perform a **minimal revision** (Line 208) to ensure high content overlap; 2. The core entity and context remain the same.
> 2) Equation 4 is **correct**. $sim()$ is a similarity metric to filter out unrelated or nonsensical generations with low similarity. A "less than" inequality would do the opposite.
>
> This ensures the generated sample is a *coherent variation*, not a random output.
>
> **Q3** Threshold for causal consistency check
>
> **RQ3** The main reason for a non-zero threshold is that the current graph G(t) used for the check **is not ground truth**.
>
> 1) G(t), derived from observational data, may be partially incorrect. Thus, this threshold acts as a **trade-off** between injecting MLLM’s knowledge and maintaining consistency with a potentially improvable graph.
> 2) Empirically, when observational data is limited or noisy, a strict zero-equality check can be overly punitive (Appendix E.8, Page 25). With the proper threshold, the cumulative effect of major consistent counterfactuals outweighs minor imperfect ones.
>
> We will clarify this in the final version.
>
> **Q4** What if MLLM has limited information
>
> **RQ4** This is a more challenging case, requiring cross-domain or domain-specific knowledge, which is currently beyond the main focus of this work. However, it can be addressed with easy extensions:
>
> 1) Set stricter validation checks to filter out noisy samples, as they are likely to violate causal constraints.
> 2) Integrate with strategies like RAG or MoE [6] to enhance fidelity in domain-specific cases.
>
> We will add a discussion of this potential extension to the paper.
>
> **References**
>
> [1] A Vashishtha, et al. Causal order: The key to leveraging imperfect experts in causal inference. ICLR, 2025
>
> [2] T Ban, et al. LLM-driven causal discovery via harmonized prior. IEEE TKDE, 2025
>
> [3] B Schölkopf, et al. Toward causal representation learning. IEEE, 2021
>
> [4] R Tu, et al. Neuropathic pain diagnosis simulator for causal discovery algorithm evaluation. NeurIPS, 2019
>
> [5] C Liu, et al. Discovery of the hidden world with large language models. NeurIPS, 2024
>
> [6] C Shi, et al. Unchosen experts can contribute too: Unleashing moe models power by self-contrast. NeurIPS, 2024

---

> > ### Author Response · Authors · 2025-08-06
> > **Rebuttal follow-up**
> >
> > Dear Reviewer yjmf,
> >
> > We sincerely appreciate your detailed review and efforts. We are kindly wondering whether our responses have adequately addressed your concerns. If you have any further feedback, we would be grateful for the opportunity to respond accordingly.
> >
> > Thank you once again for your time.
> >
> > Best regards,
> >
> > The Authors

---

> > ### Comment · Reviewer_yjmf · 2025-08-07
> > **Response to authors**
> >
> > I thank the authors for their response.
> >
> > The theoretical results makes assumptions (D.2 and D.6) that effectively proves the theorems. Whether the assumptions themselves are justified (empirically) should be the main aim of the paper. For a purely empirical paper, I don't believe enough experiments have been done in varying conditions. For example, it's still unclear to me if concepts are given to the MLLM that it does not have a clear "world model" for, will it remain uncertain or simply generate spurious samples? Further, do we know that the MLLM doesn't already know the ground truth due to data leakage?
> >
> > I'm still unsure how MCR differs from hard intervention based counterfactual? Setting a value of $V_a$ delinks it from its parents.
> >
> > With regard to the semantic similarity, again this will change a lot if the factor being "set" ($V_a=v_a$) is a parent of of the rest of the variables in the graph. For example in figure 21, if changing the nutritional profile affected the colour and shape (and other factors) of the apple, I would expect the embeddings to not be similar at all.

---

> > > ### Author Response · Authors · 2025-08-08
> > >
> > > Dear Reviewer yjmf,
> > >
> > > Thank you for your continued detailed feedback. We now provide a point-by-point response to your questions.
> > >
> > > **Q1** The theoretical results...
> > >
> > > **R1** We clarify the following:
> > > 1) We actually did not propose any theorems.
> > > 2) Justifying assumptions (as provided in rebuttal RW1, RW2.1 and RW2.2) is important, but it is certainly **not the main aim** of the paper. Instead, the main goal is to address the new problem of causal discovery from multimodal unstructured data.
> > > 3) Our work is not purely empirical. As you may have noticed, it also includes a set of theoretical discussions in Sections D and E.9.
> > > 4) We provide experiments on additional settings (i.e., simpler cases with variables defined) in rebuttal RW1, as suggested. We would highly appreciate it if you could clarify what the other conditions are.
> > > 5) As far as we know, there is currently no established method to verify if an MLLM has a clear world model. We would appreciate it if you could suggest an approach for testing it.
> > > 6) We do not ask MLLMs for ground truth but use their general knowledge to improve causal discovery, as done in [1, 2]. This should not be considered as data leakage; otherwise, the same concern would apply to all prior works using LLMs for various tasks, including causal discovery [1, 2]. Moreover, the datasets we used are newly constructed, eliminating the possibility of data leakage.
> > > 7) If we understand your meaning correctly, using an MLLM without a "world model" caused the concern on "generating spurious samples", while using one with knowledge led to "data leakage" concerns. This apparent contradiction is confusing. Highly appreciate it if you could advise an appropriate condition.
> > >
> > > **Q2** How MCR differs from hard intervention...
> > >
> > > **R2** To better elaborate, we use the same example you mentioned earlier, where the true causal relationship is $X->Y$, e.g., "nutrition"->"recmd".
> > >
> > > A hard intervention is usually applied to **only one factor in a known causal graph**; e.g., $do(Y=y')$ breaks the link to its parent.
> > >
> > > However, MCR operates in different cases where the causal direction is **uncertain**, e.g., $X-Y$. Since both variables are associated with ambiguous relationships, MCR will explore counterfactual cases for these **two factors** (X and Y) in turn, not just for Y. For example, thinking about the case "*if the factor X (nutrition) were different (lower), how would other factors change*" yields samples (e.g., x'=low nutrition, y'=low recmd) that reinforce $X->Y$. Meanwhile, by prompting the MLLM to generate counterfactual evidence to support the correct direction, it focuses more on counterfactuals aligned with $X->Y$ rather than the reverse. We will further clarify this in the paper.
> > >
> > > **Q3** With regard to the semantic similarity...
> > >
> > > **R3** We clarify that:
> > > 1. Counterfactual revisions only change certain factors without altering the context or topic, and thus do not lead to significant changes in semantic similarity.
> > > 2. We use a reasonable threshold of around 70% similarity to tolerate counterfactual variations while filtering out hallucinations.
> > >
> > > To illustrate these, we analyze several example cases and compute their similarity with original samples.
> > >
> > > > Case 1
> > > >
> > > > Original: This apple boasts a remarkable nutritional profile ... significant market potential ...
> > > >
> > > > Counterfactual (CF): This apple presents a concerning nutritional profile ... limited market potential ...
> > > >
> > > > Hallucination (HA): The city skyline shimmered under the ...
> > > >
> > > > Case 2
> > > >
> > > > Original: This apple ... a captivating aroma ...
> > > >
> > > > CF: This apple ... a concerning aroma ...
> > > >
> > > > HA: Quantum entanglement allows ...
> > > >
> > > > Case 3
> > > >
> > > > Original: This apple ... notably dry ...
> > > >
> > > > CF: This apple ... pleasingly moist ...
> > > >
> > > > HA: The towel is a rectangular piece ...
> > >
> > > Table 3.1 Comparison of semantic similarity.
> > >
> > > |Case|CF|HA|
> > > |-|-|-|
> > > |1|0.93|0.67|
> > > |2|0.94|0.69|
> > > |3|0.92|0.69|
> > >
> > > Although several factors have changed in counterfactuals, the overall semantic similarity between them and the original one remains high. However, hallucinated samples show a significant drop. This supports the use of semantic similarity to filter potential hallucinations.
> > >
> > > Recent research [3, 4] also provides strong evidence for this. Specifically, Fig. 2 in [4] shows that hallucination distributions (mostly around 50% similarity) differ from non-hallucinated ones (mostly around 80% similarity). This further validates the design of a semantic plausibility check for hallucination filtering.
> > >
> > > **References**
> > >
> > > [1] E Kiciman et al. Causal reasoning and large language models: Opening a new frontier for causality. TMLR, 2023
> > >
> > > [2] A Vashishtha et al. Causal order: The key to leveraging imperfect experts in causal inference. ICLR, 2025
> > >
> > > [3] S Farquhar et al. Detecting hallucinations in large language models using semantic entropy. Nature, 2024
> > >
> > > [4] H Oh et al. Vision-Encoders Know What They See: Mitigating Object Hallucination via Simple Fine-Grained CLIPScore. 2025

---

> ### Comment · Reviewer_yjmf · 2025-08-09
> **Response to authors**
>
> > Theoretical discussions
>
> I think the assumptions made weaken the impact of the discussed theory. Hence I think the burden of proof for the technique lies in the empirical results. For example, assumption D.2 states that the F1 (of node recovery) from the proposed CFD is greater or equal than the F1 from direct prompting. The result of Prop. 3.2 then that CFD provides better structural accuracy than direct prompting is obvious and relies heavily on assumption D.2.
>
> Similar comments hold for proposition 3.3. Assumption D.6 states that there exists $p_{MCR} >0$ for generating a counterfactual sample, which is then used to show that the graph recovery is improved with a positive probability. It's not clear to me how big this probability is, when a counterfactually inconsistent sample is generated how that hurts performance, and under what cases $p_{MCR} =0$. If we’re not sure the LLM has a coherent world model (as you state) why would we expect it to generate valid counterfactual samples?
>
> I believe this paper could be improved without these theoretical results, or a much a larger discussion of when these assumptions hold and do not hold, and why they are justified (as you did for D.2 in your response).
>
> > Experiments
>
> I would expect a much more comprehensive evaluation of the CFR module on datasets where the nodes are given. Further, evaluation to understand what the LLM does when it does not have underlying domain knowledge.
>
> > If we understand your meaning correctly, using an MLLM without a "world model" caused the concern on "generating spurious samples", while using one with knowledge led to "data leakage" concerns. This apparent contradiction is confusing. Highly appreciate it if you could advise an appropriate condition.
>
> To clarify, by this I meant whether it has memorised the causal structure due to its presence in training data [1], or whether it is actually constructing a causal model from knowledge of disparate sources. Related to my comment above, it's not clear to me if the LLM does not have detailed knowledge (causal model) of the concepts, whether it will remain uncertain or output samples that do not correspond to a random casual structure.
>
> [1] Bordt, Sebastian, et al. "Elephants never forget: Memorization and learning of tabular data in large language models." arXiv preprint arXiv:2404.06209 (2024).
>
> >  Counterfactual samples
>
> I'm not quite understanding how your counterfactual samples are generated? In your example, if you explore **both counterfactuals**, if you ask it to generate samples from "if the factor Y (recmd) were different (lower), how would other factors change", this will break the link between X and Y and generate independent samples (see [Ch.4, 2]). Feeding this into algorithms like PC can hurt results.
>
> [2] Pearl, Judea, Madelyn Glymour, and Nicholas P. Jewell. Causal inference in statistics: A primer. John Wiley & Sons, 2016.
>
> Due to lack of time in the response period, I'm happy to raise my score based on the response so far.

---

> > ### Author Response · Authors · 2025-08-09
> >
> > Dear Reviewer yjmf,
> >
> > We greatly appreciate your further feedback and thank you for considering raising the score. We would be glad to discuss the details with you further if more time were available. We will further clarify these points you mentioned in the final version of our paper.
> >
> > Best regards,
> >
> > The Authors

---

### Official Review · Reviewer_p1ec · 2025-07-05

**Clarity:** 2
**Significance:** 2
**Originality:** 2
**Rating:** 4
**Confidence:** 4

**Summary:**

The paper claims to solve the problem of latent variable causal discovery using a multi-modal foundation modeling framework. It uses CLIP embeddings from multi-modal inputs using contrasting learning, and then performs FCI on these embeddings. It further refines this graph by generating counterfactual samples and reintegrating them into the training loop.

**Questions:**

1. Why are the optimal graphs in proposition 3.2 different?
2. In the proof of 3.2, how do authors justify making Assumption D.2? What does it mean to use a direct approach to obtain the features -- could the authors be more specific about these techniques?
3. How do the authors solve FCI within a continuous optimization framework? Furthermore, they claim that their algorithm can solve the latent variable problem but the ground truth graph does not include any latent variables. Could the authors elaborate more on that?

**Ethical Concerns:**

["NO or VERY MINOR ethics concerns only"]

**Limitations:**

Yes

**Quality:**

2

**Strengths And Weaknesses:**

Strengths:
- The overall idea to use CLIP and causal graph on top of it innovative.
- The empirical results look strong, although the causal graphs being used seems simpler than how general the authors claim that their algorithm is, in the paper.

Weaknesses:
- Lack of clarity:
1. \Psi is used before it is defined.
2. Counterfactual generation and how it is done is not entirely clear and could be better described in the text.
3. It is not clear how FCI was optimized within an LLM framework, since it's a constraint-based algorithm.
4. There are some concerns regarding the theory - please see questions below.
5. The empirical results do not include ground truth graphs with latent variables (when in the space of factors), yet the authors claim that their method can successfully work in this setting.

---

> ### Author Rebuttal · Authors · 2025-07-31
>
> We are grateful for the reviewer's constructive comments, which have helped us significantly improve the clarity of our paper.
>
> **W1.** $\Psi$ is used before it is defined.
>
> **RW1.** We thank the reviewer for pointing this out. The symbol $\Psi$ is first defined and used on Page 4, Line 155, where we state “This module guides the MLLM $\Psi$ to explore the intra- and inter-modal interactions…”. Here, $\Psi$ is defined as the MLLM for contrastive factor discovery and multimodal counterfactual reasoning. It is also defined in our Appendix A (Table of Notations, Page 2). To improve clarity, we will define it more explicitly and clearly in Line 155, as “This module adopts an MLLM, denoted as $\Psi$, to explore the intra- and inter-modal interactions…”
>
>
> **W2.** Counterfactual generation and how it is done is not entirely clear and could be better described in the text.
>
> **RW2.** Thank you. The process of counterfactual generation is introduced in Section 3.4 (Page 5), with other details that can help comprehension, such as the organized algorithm in Appendix C (Page 5), and illustrated examples in Appendix E.10 (Figures 21&22, Page 26). Specifically, the process includes:
>
> 1) **Identifying Uncertainty**: We select factors with ambiguous relationships (e.g., undirected edges) in the computed graph $G^{(t)}$.
> 2) **Prompting for Reasoning**: We use the MLLM ($\Psi$) with a “what-if” prompt to reason about the counterfactual scenario of an intervention on a candidate factor (e.g., In the given sample, if the factor “nutrition” were different, how would other factors be affected?)
> 3) **Generating Samples**: By thinking of the counterfactual value of each factor, model $\Psi$ generates the corresponding multimodal counterfactual samples. Textual data can be directly revised by model $\Psi$ from the original text to reflect the changes in factor values, while visual data will be produced by an image generation model $\Phi$ with a brief description of image modifications by model $\Psi$. Some generated examples can be seen in Figures 21&22, Page 26.
> 4) **Validation**: We validate these generated samples for semantic plausibility and causal consistency to filter out incorrect and implausible counterfactuals.
>
> Thank you for your suggestions. We will revise Section 3.4 to make this step-by-step process more explicit.
>
>
> **W3.** It is not clear how FCI was optimized within an LLM framework, since it's a constraint-based algorithm.
>
> **RW3.** Thank you. There may be some misunderstanding. Actually, we do not optimize FCI within an LLM framework. Our method, MLLM-CD, is a three-stage process:
>
> 1) **Factor Discovery (LLM-based)**: We use MLLMs to transform unstructured multimodal data into a structured dataset with causal factors and their values (as examples in Table 3.1, and an illustrative example in Figure 23, Page 27).
> 2) **Causal Structure Discovery (Statistical)**: We then apply a **standard statistical** causal discovery algorithm like FCI to learn causal structures from the extracted structured data.
> 3) **Counterfactual Reasoning (LLM-based)**: We use MLLM’s reasoning ability to generate plausible counterfactual samples for those factors with ambiguity to iteratively refine the discovery process.
>
> As stated in Appendix E.3.1 (Page 16, Line 292), we use the FCI implementation from the causal-learn library. The LLM does not interact with the internal workings of FCI to maintain the **statistical rigor in causal discovery**. This hybrid paradigm is also used and verified effective in relevant work [1]. We will clarify this in the revised paper.
>
> **Table 3.1** Examples of extracted structured data.
> |nutrition|color|defects|aroma|taste|juciness|sentimen|recmd|score|
> |:-:|:-:|:-:|:-:|:-:|:-:|:-:|:-:|:-:|
> |-1|1|1|1|-1|1|1|1|3|
> |-1|0|-1|-1|-1|1|-1|-1|-5|
> |...|...|...|...|...|...|...|...|...|
>
>
> **W4**. There are some concerns regarding the theory - please see questions below.
>
> **RW4.** Thanks. We have provided the point-by-point responses in **RQ1** and **RQ2** to answer the questions regarding theories. Hope they address your concerns.
>
>
> **W5.** The empirical results do not include ground truth graphs with latent variables (when in the space of factors), yet the authors claim that their method can successfully work in this setting.
>
> **RW5.** Thank you for your comments. We did not claim addressing “latent variables” as a contribution of our work. We mean those algorithms robust to latent variables are relevant to addressing this setting with latent variables.
>
> We only mentioned “latent variables” in Lines 183-184: “*Given the potential presence of unobserved confounders in real-world scenarios, algorithms robust to latent variables, such as FCI algorithm [29] or its variants (e.g., RFCI [64]), are particularly relevant.*” Here, “latent variables” refer to unobserved confounders, which make causal structure discovery more challenging. Since FCI is known to be more robust to confounders in real-world applications [2, 3], we choose it as the causal structure discovery algorithm in our framework. Thus, the ability to handle latent variables is a merit of the FCI algorithm itself and is beyond our contribution. We will clarify this in the paper and explore this complex setting in future work.
>
>
> **Q1.** Why are the optimal graphs in proposition 3.2 different?
>
> **RQ1.** The optimal graphs $G^\*\_{V\_{CFD}}$ and $G^\*\_{V\_{direct}}$ in proposition 3.2 differ from the notation $G^*$ (Line 236) because, given two subsets of factors discovered by two methods, the performance is also compared based on the corresponding optimal subgraphs induced by these factor subsets (as defined in Lines 242-243).
>
>
> **Q2.** In the proof of 3.2, how do authors justify making Assumption D.2? What does it mean to use a direct approach to obtain the features -- could the authors be more specific about these techniques?
>
> **RQ2.** The **"direct approach"**, as you may notice, is defined in Lines 74-75, Page 6. It refers to a non-contrastive baseline where an MLLM is directly prompted to identify causal factors from data samples, e.g., by asking "*what are the high-level factors that contribute to the apple scores*", similar to the baseline method **COAT [1]**. We will provide the direct prompt in Appendix E.2 for clarity.
>
> **Justification for Assumption D.2:** We would like to justify making Assumption D.2 from the following two aspects:
>
> 1) **Mechanistic Design:** Our contrastive factor discovery (CFD) is superior to direct prompting as it offers critical guidance to uncover factors hidden in multimodal unstructured data. CFD forces the MLLM to focus on salient differences in cross-modal and within-modal pairs of samples, guiding it to identify genuine causal factors rather than superficial features. The direct approach, such as the one used in COAT, lacks this guidance and thus usually fails to discover comprehensive factors from multimodal data, as seen in Figure 2 (Page 2).
> 2) **Empirical Evidence:** Our experimental results provide strong evidence for this assumption: **Tables 1&2** (Page 7, 9) show that MLLM-CD (using CFD) consistently achieves significantly higher **Node Recall (NR)** and **Node F1 (NF)** for factor discovery compared to the direct approach of COAT. A similar observation can also be found in the ablation study in Table 3 (Page 9), where the NF metric significantly decreases by 36% on the Lung dataset without CFD (comparing the rows of *w/o CFD* and *MLLM-CD*).
>
> Additionally, to further enhance the empirical justification, we now provide a direct comparison between CFD and direct prompting on the performance of the factor proposal. Key results are in the following Table 2.1. As we can see, CFD consistently outperforms direct prompting in terms of correctness and completeness, and the significance test results (p<0.05) indicate that this large improvement is statistically meaningful. Thus, Assumption D.2 can be empirically justified.
>
> **Table 2.1** Comparison between CFD and direct prompting on the factor discovery performance.
>
> |      |         | NP↑         | NR↑         | NF↑         |
> | ---- | ------- | ---------- | ---------- | ---------- |
> | MAG  | Direct  | 0.6945     | 0.4074     | 0.5094     |
> |      | CFD     | **0.8796** | **0.8148** | **0.8453** |
> |      | p-value | 0.0129     | 0.0041     | 0.0023     |
> | Lung | Direct  | 0.6667     | 0.5333     | 0.5926     |
> |      | CFD     | **0.8889** | **1.0000** | **0.9394** |
> |      | p-value | 0.0471     | 0.0099     | 0.0165     |
>
>
> **Q3.** How do the authors solve FCI within a continuous optimization framework? Furthermore, they claim that their algorithm can solve the latent variable problem but the ground truth graph does not include any latent variables. Could the authors elaborate more on that?
>
> **RQ3.** Thank you. As we explained in **RW3**, we apply the standard FCI algorithm to the extracted factors and structured data to compute causal structures. The LLM does not interact with the internal workings of FCI to maintain the **statistical rigor in causal discovery**.
>
> In addition, as clarified in **RW5**, solving latent variables is not a main focus or contribution of our work. The term “latent variables” appears only in Lines 183-184 to introduce an inherent strength of FCI, not to position it as the focus of this work. We will clarify this point in Section 3.3. Thanks to your comments, we will also consider exploring more complex and challenging scenarios involving latent variables in future work.
>
>
> **References:**
>
> [1] C Liu, et al. Discovery of the hidden world with large language models. NeurIPS, 2024.
>
> [2] P Spirtes, C N Glymour, R Scheines. Causation, prediction, and search. MIT Press, 2000.
>
> [3] J M Ogarrio, P Spirtes, J Ramsey. A hybrid causal search algorithm for latent variable models. Conference on Probabilistic Graphical Models. 2016.

---

> > ### Author Response · Authors · 2025-08-06
> > **Rebuttal follow-up**
> >
> > Dear Reviewer p1ec,
> >
> > We sincerely appreciate your detailed review and efforts. We are kindly wondering whether our responses have adequately addressed your concerns. If you have any further feedback, we would be grateful for the opportunity to respond accordingly.
> >
> > Thank you once again for your time.
> >
> > Best regards,
> >
> > The Authors

---

> > ### Author Response · Authors · 2025-08-09
> > **A gentle reminder for your feedback before the close of the discussion**
> >
> > Dear Reviewer p1ec,
> >
> > We sincerely appreciate your detailed review and efforts. As the discussion deadline is approaching, with less than 10 hours remaining, we would like to kindly ask whether our responses have addressed your concerns. We would be grateful for any further feedback so that we may have time to respond accordingly.
> >
> > Thank you once again for your time and consideration.
> >
> > Best regards,
> > The Authors

---

> > > ### Author Response · Authors · 2025-08-09
> > >
> > > Dear Reviewer p1ec,
> > >
> > > As the discussion period will close soon, we are keen to know if your questions and concerns have been addressed by our response. Thank you so much.
> > >
> > > Best regards,
> > >
> > > The Authors

---

> > > > ### Comment · Reviewer_p1ec · 2025-08-09
> > > >
> > > > Dear Authors,
> > > >
> > > > Thank you for your detailed response. Some of my concerns have been addressed, however I do agree with Reviewer yjmf that this paper could use expanding on empirical results as well as discussion on the assumptions. Therefore, I have decided to maintain my score.

---

### Comment · Area_Chair_N5Ny · 2025-08-03
**Author–Reviewer Discussion Open: Early Participation Appreciated**

Dear Reviewers,

Please take time to read the other reviews and author responses carefully, and actively participate in the Author–Reviewer Discussions—posting an initial comment early, even a brief one, helps enable a constructive exchange. Thank you!

---

### Note · Authors · 2025-08-12

Dear Reviewers, ACs, SACs, and PCs,

We sincerely thank you for your constructive feedback and efforts in handling our paper. During the rebuttal period, we incorporated additional empirical justification and discussion to enhance the quality and clarity of our work. A brief summary follows:

**Core contribution**
* **Significance** (Reviewers ynPz, yjmf): To the best of our knowledge, this is the first study to leverage MLLM knowledge to extend the scope of causal discovery (CD) to **multimodal unstructured data**. This direction is largely underexplored (with only one prior study), yet holds **significant practical value**, with potential to advance scientific progress in various domains. For example, purchase behavior modeling in consumer economics (MAG dataset), disease diagnosis in healthcare (Lung dataset), habitat assessment in ecological monitoring, and molecular interaction studies in chemoinformatics.
* **Originality** (p1ec, EBFe, ynPz): Two novel designs ensure its effectiveness:
  * **Contrastive factor discovery (CFD)** identifies more accurate and comprehensive factors using intra- and inter-modal contrastive signals.
  * **Multimodal counterfactual reasoning (MCR)** generates plausible, consistency-checked counterfactuals using MLLM general knowledge to alleviate structural ambiguity.
* **Strong Empirical Validation** (p1ec, EBFe, ynPz): We build the first two datasets for this task, one synthetic (MAG dataset) and one real-world (Lung dataset). Extensive experiments show the superior performance over SOTA baselines and the effectiveness of each component.

**Revisions made based on rebuttal discussions**
* Comparisons between CFD and direct (non-contrastive) methods to support **Assumption D.2**;
* Quantitative and qualitative results on counterfactuals to support **Assumption D.6**, with discussion on: 1) examples showing $p_{MCR}>0$, 2) validation checks to filter low-quality generations, and 3) extensions for domain-specific cases;
* **Validation of CFD and MCR modules** in traditional settings with defined variables; implementation details of MCR to show its mechanism and how it focuses on **causally consistent** rather than **independent generations**;
* Clarification of validation checks, the role of FCI, and future directions with latent variables and domain-specific knowledge; discussion of connections and differences with causal representation learning.
* Ablation study on CFD **sampling strategies**.

Best regards,

The Authors

---

### Decision · Program_Chairs · 2025-09-17

**Decision:**

Accept (poster)

**Comment:**

This paper presents MLLM-CD, a framework for causal discovery from unstructured multimodal data, combining Contrastive Factor Discovery, causal structure learning, and Multimodal Counterfactual Reasoning. Reviewers noted that the paper is well-motivated, technically coherent, and demonstrates strong empirical performance on synthetic and real-world datasets. The use of MLLMs for factor discovery and counterfactual generation is a novel and appealing aspect. While some concerns remain regarding theoretical assumptions, limited ablations, and dataset scale, these do not outweigh the paper’s strengths. Overall, the work makes a meaningful contribution to multimodal causal discovery.